

# Extreme precipitation associated with atmospheric rivers over West Antarctic ice shelves: insights from kilometre-scale regional climate modelling

Ella Gilbert[1], Denis Pishniak[2], José Abraham Torres[3], Andrew Orr[1], Michelle Maclennan[4], Nander Wever[5], Kristiina Verro[6]

[1]British Antarctic Survey, Madingley Road, Cambridge, UK.
[2] National Antarctic Science Centre of Ukraine, Kyiv, Ukraine.
[3] Danish Meteorological Institute, Copenhagen, Denmark.
[4] University of Colorado Boulder, Boulder CO, USA.
[5] WSL Institute for Snow and Avalanche Research SLF, Davos, Switzerland.
[6] Institute for Marine and Atmospheric Research, Utrecht University, Utrecht, Netherlands.

*Correspondence to*: Ella Gilbert (ellgil82@bas.ac.uk)

**Abstract.** We explore how atmospheric rivers (ARs) in a summer and a winter case interact with the topography of the Amundsen Sea Embayment, West Antarctica, and deposit significant amounts of precipitation. To do this we use results from three regional climate models (RCMs: MetUM, Polar-WRF, HCLIM) at a spatial resolution of 1 km. Estimates of snowfall associated with both these events from all three RCM simulations compare well against observed snow height measurements over the Thwaites and Pine Island ice shelves. By contrast, snowfall estimates from ERA5 reanalysis for both events are severely underestimated compared to the measurements. Outputs from the RCMs also show that the ARs may be associated with several millimeters of rain in both the summer and winter cases, although in the absence of *in situ* measurements of rainfall, this result cannot be directly verified. The RCM simulations suggest that the rainfall during these events can fall directly as supercooled drizzle, but also that rainfall is concentrated around steep terrain due to the interaction of ARs with complex orography. We also show that while the amount of RCM-simulated snowfall was comparatively resolution insensitive, the amount of rainfall simulated was not, with rainfall amounts much higher in 1 km simulations compared to 12 km simulations. Our work highlights that kilometer-scale models are useful tools to investigate the total precipitation amount and its partitioning into rain and snow over this globally important and climatically sensitive region, as well as the critical need for *in situ* observations of rainfall.

# 1 Introduction

West Antarctica, and particularly the Amundsen Sea Embayment (ASE), has been the focus of attention in recent years because of the rapid pace of climate and cryospheric change there. Glaciers in the ASE, and in particular the Thwaites (TG) and Pine Island (PIG) glaciers, have been highlighted because they are accelerating and losing ice mass extremely fast, largely due to basal melting (Lhermitte et al., 2020; Rignot et al., 2019). The ice shelves restraining TG and PIG have been shown to be



vulnerable to damage and weakening, and are changing very rapidly (Lhermitte et al., 2020; Alley et al., 2021). In fact, over a third of the mass loss from West Antarctica comes from ice shelves (Smith et al., 2020).

The health of ice shelves can be partly measured by the surface mass balance (SMB). Ice shelf SMB describes how inputs of precipitation (accumulation) are balanced by losses of melt runoff, sublimation and evaporation (ablation). Snowfall accumulation is the primary counterbalance to dynamical and surface losses from ice shelves, with redistribution (e.g. via blowing snow) playing an additional role (Mottram et al. 2020; Lenaerts et al., 2019). Accumulation does not currently offset the aforementioned losses on ASE ice shelves (Rignot et al., 2019) but is vital to constrain because it can have an important

effect on the timing and characteristics of ice shelf collapse, or recession and thinning (Scambos et al., 2017; Medley & Thomas, 2019; van Wessem et al., 2023). Antarctic accumulation is expected to increase as climate changes because a warmer atmosphere has a higher saturation vapour capacity as per the Clausius-Clapeyron relationship (Clausius, 1850; Clapeyron, 1834), meaning it can retain more moisture. However, increased precipitation will likely not balance increased losses over the 21$^{st}$ century (Gilbert & Kittel, 2021; Kittel et al., 2021).

Extreme precipitation events account for a large proportion of annual precipitation on the ASE coast, with 50% of annual precipitation falling in approximately 30 days per year (Turner et al., 2019). And they matter: Davison et al. (2023) demonstrate that extreme precipitation can play an important role in offsetting some of the notable mass losses in the ASE, and Wille et al. (2024) show that extreme precipitation in East Antarctica in March 2022 helped to make 2022's annual mass balance positive – a rare occurrence.

Moreover, atmospheric rivers (ARs) often coincide with extreme precipitation events in the ASE region and account for up to 11 % of annual precipitation totals, despite making landfall on just ~3 days per year (Wille et al., 2021, Maclennan et al., 2023). These long, filament-shaped atmospheric features transport large quantities of water vapour meridionally, and have been described as "rivers in the sky" that can greatly impact Antarctic precipitation variability (Shields et al., 2022). ARs are frequently associated with the presence of a high/low pressure couplet and blocking anti-cyclone that funnels warm and moist

air towards the continent (Wille et al., 2019; Scott et al., 2019; Baiman et al., 2024). Adusumilli et al. (2021) showed that extreme precipitation events explained 41% of snow height changes over the West Antarctic ice sheet in 2019 and that the majority were associated with the infrequent but intense precipitation brought by ARs. However, Davison et al. (2023) note that 2019 and 2020 were both extreme precipitation years, implying that these results may not be representative.

The localised impacts of ARs in Antarctica have been shown to be extremely important. The characteristics of ARs in

Antarctica differ from those elsewhere – for instance they make landfall less frequently and the colder polar atmosphere means they transport less moisture than at lower latitudes (Maclennan et al., 2023). They are also strongly impacted by the steep terrain of the Antarctic ice sheet, which often results in the coincidence of ARs with foehn events, raising temperatures and wind speeds but causing a drying (Elvidge & Renfrew, 2016; Gilbert et al., 2022). This foehn + AR combination has previously



been shown to be important over the Antarctic Peninsula (Bozkurt et al., 2018; Wille et al., 2022; Zou et al., 2023;
Gorodetskaya et al., 2023) and South Orkney Islands (Lu et al., 2023). Francis et al. (2023) also show that foehn winds over
PIG have a considerable impact on the ice shelf mass balance via their effect on sublimation and that these can be combined
with ARs to induce considerable warming. But they can also impact precipitation, for instance Gehring et al. (2022) show how
an AR and foehn event combined to affect precipitation phase and the quantity of snow reaching the ground at Davis, East
Antarctica.

ARs are also associated with the occurrence of rainfall over Antarctica (Wille et al., 2019, 2021, 2024; Lu et al., 2023). For
example, ARs were responsible for approximately 30% of the rainfall recorded in the ASE between 1980-2018 (Wille et al.,
2021) and were implicated in the record-breaking East Antarctic heatwave of March 2022, which resulted in rainfall over the
periphery of the ice sheet and ice shelves (Wille et al., 2024). Rainfall is an important climate indicator but can also contribute
to surface melting by the freezing of the liquid water within the snowpack releasing latent energy to the environment, as well
as lowering the surface albedo (Doyle et al., 2015; Wille et al., 2021; Box et al., 2022; 2023). Rainfall also impacts firn
properties by filling pore spaces and driving densification, and if it refreezes in the snowpack rain can form ice lenses that
reduce the stability of ice shelves (Harper et al. 2023; Noël et al., 2023). The interaction of airflow with steep terrain can also
impact the phase and characteristics of precipitation, for example enhancing rainfall production via the seeder-feeder
mechanism (Lean & Clark, 2003) and generating supercooled drizzle in mixed-phase clouds (Fernandez-Gonzalez et al., 2015;
Ramelli et al., 2021).

The quantity and characteristics of precipitation falling over ice shelves in the ASE are highly dependent on localised
characteristics such as orography, which are not adequately represented by either sparse *in situ* observations or coarse
resolution atmospheric reanalyses (Turner et al., 2019; Mottram et al., 2020; Gehring et al., 2022; Nicola et al., 2023). This
constitutes a clear knowledge gap, which must be addressed to better estimate SMB and loss of mass from ice sheets (IPCC,
2019; Pritchard, 2021). Additionally, while several studies have used reanalysis to explore precipitation extremes associated
with ARs over the ASE region (e.g. Francis et al., 2023; Maclennan et al., 2023; Wille et al., 2021), such datasets do not have
sufficient resolution to explore the impact of complex orography on AR precipitation dynamics and characteristics. Using
regional climate models (RCMs) to dynamically downscale reanalysis to high spatial and temporal resolution has been shown
to be critical to resolve the important fine-scale processes, interactions, and circulations that contribute to precipitation,
including extremes (Gilbert et al., 2020; 2022; Morrison et al., 2020; Lu et al., 2023).

This study will use a mini-ensemble of kilometre-scale RCMs to explore how ARs over the ASE interact with the TG and PIG
ice shelves in two case studies, and deposit significant amounts of precipitation, including as rain. We will also evaluate the
performance of these RCMs in this region during extreme precipitation conditions. This work will help us understand the



drivers and impacts of extreme precipitation associated with ARs, and therefore to understand to what extent ice mass loss
from important ice shelves and glaciers in the region can be modulated by accumulation.

## 2 Materials and methods

We use a mini-ensemble of three regional climate models (RCMs: MetUM, Polar-WRF and HCLIM) over the region centred
on the PIG and TG ice shelves to simulate two extreme precipitation / AR case studies: a winter case (23-30 June 2020) and a
summer case (3-9 February 2020), which is also used in Maclennan et al. (2023). We then compare simulation outputs with
near-surface meteorological automatic weather station (AWS) data on the TG ice shelf, as well as accumulation data derived
from observations of snow height at the station to provide a ground-truth of accumulated mass. We also include simulations
by the SNOWPACK snow model (Wever, 2022), where the model derived accumulation and snow water equivalent are derived
directly from the snow height sensor measurements (Maclennan et al., 2023). SNOWPACK allows a direct comparison of the
observations against RCMs and ERA5 reanalysis (which output snow and rain mass quantities), and therefore provides a
ground-truth of accumulated mass that enables us to evaluate precipitation phase partitioning. Further information regarding
the RCMs is given in section 2.1, the ERA5 reanalysis is described in section 2.2, the details of the observational data are
given in section 2.3, and the SNOWPACK firn model is described in section 2.4.

## 2.1 Regional Climate Models

All three RCMs use a one-way nesting approach to dynamically downscale ERA5 reanalysis (Hersbach et al., 2020, described
in section 2.2) to very fine resolution (1 km horizontal grid spacing) with a series of intermediate nests. The MetUM and
HCLIM use exactly the same domains, with the 1 km inner domain nested within an intermediate 3 km and 12 km domain, as
shown in Figure 1a. Figure 1b shows the 1 km domain, which is focused on the region of interest. Meanwhile, Polar-WRF
uses a setup focused on the same region but with a 9 km outer domain, and 3 km and 1 km domains with a larger areal extent,
as shown in Figure S1. The RCMs all have a model top of approximately 40 km, with 52, 65 and 70 vertical levels in Polar-
WRF, HCLIM and MetUM, respectively, with 16, 20 and 14 levels below 1 km. All three RCMs derive their model surface
elevation and land/sea mask from updated high resolution datasets. For the MetUM and Polar-WRF this is the REMA dataset
at 200 m resolution (Howat et al., 2019), whereas for HCLIM this is v3 of the MEaSUREs BedMachine elevation model
(Morlighem et al., 2022) at 500 m resolution.

The ERA5 reanalysis is used to supply the RCMs with initial and boundary conditions. For Polar-WRF and HCLIM, these
conditions are applied directly to the edges of the outer nest (9 km and 12km, respectively). For the MetUM, ERA5 is used to
force a global model configuration of the MetUM at approximately 40 km horizontal resolution, which then provides boundary
conditions for the outermost 12 km nest of the MetUM. Both HCLIM and Polar-WRF are run continuously for each case with
boundary conditions supplied at three-hourly intervals, and the first 24 hours of the model run discarded as spin-up. The



MetUM is run in 'forecast mode', which means it is re-initialised at 12-hourly intervals, and then run for 24-hour forecasts. The initial 12 hours of each forecast are discarded as spin-up, while the second halves are kept and concatenated into a continuous time series, as in Gilbert et al. (2022).

The following sub-sections describe relevant technical differences and parameterisations used in each model.


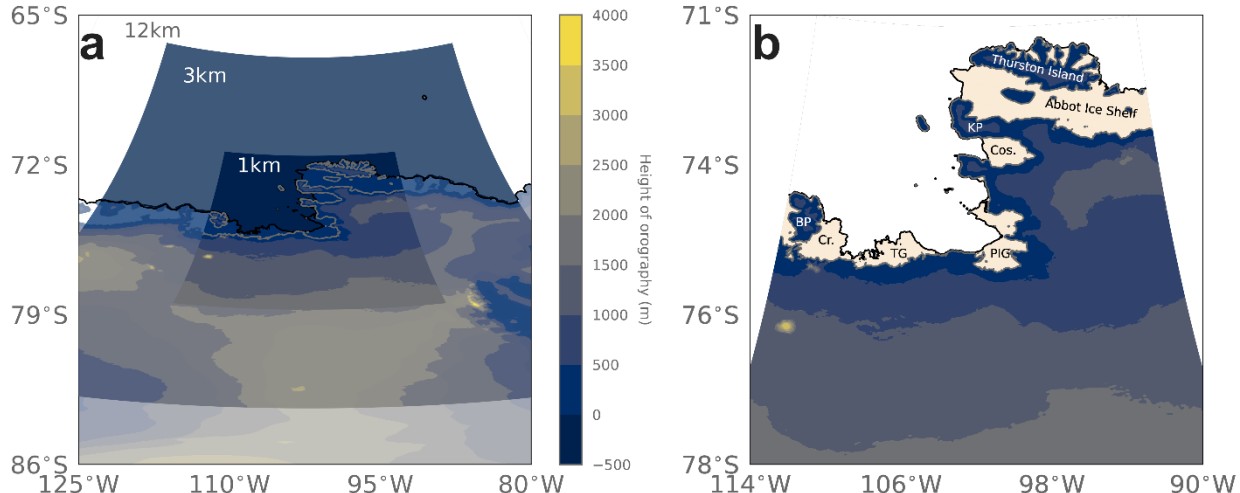

**Figure 1.** Model domains used in this study. Panel a) shows the mean height of REMA orography (Howat et al., 2019) over the 12 km, 3 km and 1 km domains, with ice shelf outlines shown in grey on the innermost 1 km domain. Panel b) shows the 1 km domain, with ice shelves shown in off-white and important locations labelled as follows: BP – Bear Peninsula, Cr. –

Crosson Ice Shelf, TG – Thwaites Glacier Ice Shelf, PIG – Pine Island Glacier Ice Shelf, Cos. – Cosgrove Ice Shelf, KP – King Peninsula, Abbot Ice Shelf, Thurston Island.

### 2.1.1 MetUM

The MetUM global model is run using the GA7 science configuration (Walters et al., 2019), while the nested domains are run using the RA2M science configuration described in Bush et al. (2023). This means that convection is parameterised in the 12

km nest (according to the 5a scheme originally developed by Cullen, 1993), while it is explicitly resolved in the 3 km and 1 km domains. The formulation of the (diagnostic) large-scale cloud scheme used in RA2M is similar to the configuration described in Gilbert et al. (2020), including the modifications of Abel et al. (2017). These adaptations limit the overlap between cloud liquid and ice phases, thereby reducing the positive cloud ice and negative cloud liquid biases present in previous model versions. The microphysics scheme is based on Wilson & Ballard (1999) with extensive modifications and represents five

hydrometeors (cloud, ice, snow, rain and graupel). The single-moment scheme uses a fixed cloud droplet number and generic



ice particle size distribution of Field et al. (2007). Further details of the scientific specification of RA2M can be found in Bush et al. (2021).

### 2.1.2 Polar-WRF

Polar-WRF is a version of the Advanced Research WRF core (Skamarock, 2019) modified to perform well in the polar regions
(Xue et al., 2022). Cloud fraction is parameterised using the diagnostic scheme of Xu & Randall (1996). Convection is parameterised in the coarsest 9 km domain using the Kain-Fritsch scheme for deep and shallow convection (Kain, 2004), and explicitly resolved in the 3 km and 1 km domains. Precipitation and cloud microphysics are parameterised according to the scheme of Morrison et al. (2008), which is a double-moment scheme that represents four hydrometeors (ice, snow, rain and graupel) and is suitable for cloud-resolving simulations. Further description of the physics configuration of Polar-WRF can be
found in Hines et al. (2019), Bromwich et al. (2013) and Wilson et al. (2011).

### 2.1.3 HCLIM

HCLIM43–AROME (HCLIM in this study) uses a statistical large-scale cloud scheme based on Bechtold et al. (1995) to determine cloud fraction, and shallow convection is parameterised according to the eddy diffusivity mass-flux framework (EDMFm; de Rooy and Siebesma, 2008; Bengtsson et al., 2017). A single-moment cloud microphysics scheme, ICE3 (Pinty
and Jabouille, 1998; Lascaux et al., 2006), is used to parameterise precipitation, with modifications applied for cold conditions (OCND2; Müller et al., 2017). The scheme represents six species of water (vapour, cloud droplets, pristine ice, rain, snow/aggregates and frozen drops/graupel) and the OCND2 modifications separate fast liquid-phase processes from the slower cold-phase processes. A more detailed description of the HCLIM-AROME model system is presented in Belušić et al. (2020).

### 2.2 ERA5 reanalysis

ERA5 is the latest reanalysis product from the European Centre for Medium Range Weather Forecasting (ECMWF), spanning the period 1940 to present. The production of ERA5 involves the assimilation of vast quantities of historical observations into the ECMWF's operational forecast model (IFS, at cycle 41r2), which outputs variables hourly on 137 vertical levels at ~31 km horizontal resolution. The model's surface orography is interpolated from numerous elevation datasets, including SRTM30, as described in ECMWF (2016). Its surface scheme, HTESSEL, represents surface fluxes, snow and SMB-relevant quantities
like melt and runoff on surface tiles (Balsamo et al., 2009), while four cloud species (cloud liquid, ice, rain and snow) and fractional cloud cover are described prognostically by the precipitation and large-scale cloud schemes (ECMWF, 2016). Radiation is computed by the McRad scheme (Morcrette et al., 2008).

### 2.3 *In situ* data description

Observational data are taken from two instrumented stations on the eastern side of the TG ice shelf: Cavity Camp (75.033°S
105.617°W) and Channel Camp (75.050°S, 105.4334°W), situated approximately 4 km apart. Cavity Camp is located on a



flat, relatively spatially homogeneous area of the ice shelf, whereas Channel Camp is located in the surface expression of a basal melt channel, which can be regions that preferentially develop surface crevasses (Alley et al., 2016; 2021). The stations are equipped with Automated Meteorology–Ice –Geophysics Observation System (AMIGOS) instruments, including sensors to measure near-surface meteorology like near-surface air temperature, relative humidity, pressure and wind speed, as well as firn temperature, snow height and GPS position. A full description of the AMIGOS setup is given in Scambos et al. (2013).

## 2.4 SNOWPACK model

We also use output from simulations using the SNOWPACK multi-layer firn model (Wever, 2022) that were performed by Maclennan et al. (2023). In these simulations, the model is used to convert the observed snow height data from the AMIGOS instruments on the TG ice shelf to mass accumulation (Lehning et al., 2002a; 2002b). The SNOWPACK model calculates the surface energy balance based on the observed near-surface meteorological variables, including wind speed, temperature, relative humidity, and incoming short- and longwave radiation. It uses the surface energy balance to diagnose surface height changes from terms like surface melt and sublimation, and also calculates snow height decreases from compaction. Then, the observed snow height is compared to the observed snow height to determine the accumulation terms. The model diagnoses snowfall when all three of the following conditions are met: observed snow height exceeds the simulated height, relative humidity exceeds 50%, and the difference between air and snow surface temperatures is less than 3°C (Lehning et al. 1999; Wever et al., 2015). Accumulated snow mass is calculated according to Schmucki et al. (2014) using the density of fresh snow, while describing the effect of snow erosion and deposition on snow density using the Redeposit scheme (Wever et al., 2023). This approach provides time-varying new snow density, which, for this area, results in a calculated average surface snow density of approximately 450 kg m$^{-3}$. Rainfall is assumed when the air temperature exceeds 1.2°C, during periods where the forcing model predicts precipitation.

## 3 Results & Discussion

### 3.1 Case description

Both the summer and winter cases examined in this work have a similar synoptic setup, with cross-terrain flow established by an atmospheric pressure dipole between the northwestern Antarctic Peninsula (high pressure) and Amundsen Sea (low pressure), as shown in Figure 2. This same synoptic setup is shown to be responsible for establishing cross-terrain flow and foehn winds over PIG and the wider ASE region (Francis et al., 2023). Both feature extreme precipitation, with SNOWPACK-simulated accumulation totals of 108 mm w.e. and 80 mm w.e. in summer at the Cavity and Channel stations, respectively and 61 mm w.e. and 113 mm w.e., respectively, during winter. In both cases, observations suggest that near-surface temperatures rose to the melting point of 0°C, supporting previous studies (e.g. Maclennan et al., 2023; Lu et al., 2023) that extreme precipitation events can have consequences not only for accumulation, but also for melting (see Figure S2).



The summertime case consists of a family of three ARs that made landfall over TG and the ASE coast in quick succession, driven by a high-low pressure couplet (Figure 2a). A pressure dipole, with low pressure over the Amundsen Sea (reaching a minimum MSLP value of 951 hPa in ERA5) and high pressure over the eastern tip of South America (maximum MSLP

of up to 1035 hPa), drove convergence along a pressure ridge that extended lengthways along the eastern side of the Antarctic Peninsula. The high pressure system strengthened over the course of the case, enhancing the pressure gradient and driving more intensified convergence. This channelled moist air towards and across the West Antarctic coast, especially over the region of TG / PIG, and prevented the low pressure systems over the Amundsen Sea from migrating. The air mass picked up moisture as it travelled over the largely sea ice-free Bellingshausen Sea west of the peninsula and was forced

over the topography of the West Antarctic Ice Sheet.

The wintertime case consists of high pressure over the Drake Passage and Bellingshausen Sea (top left of Figure 2b, with ERA5 MSLP reaching up to 1035 hPa) and low pressure over the Amundsen Sea (bottom left of Figure 2b, with ERA5 MSLP falling as low as 955 hPa) (Figure 2b). A chain of low pressure systems pushed towards the ice sheet from sub-polar

latitudes all the way to approximately 85°S during the case, driving high temperatures and precipitation over the ASE region.





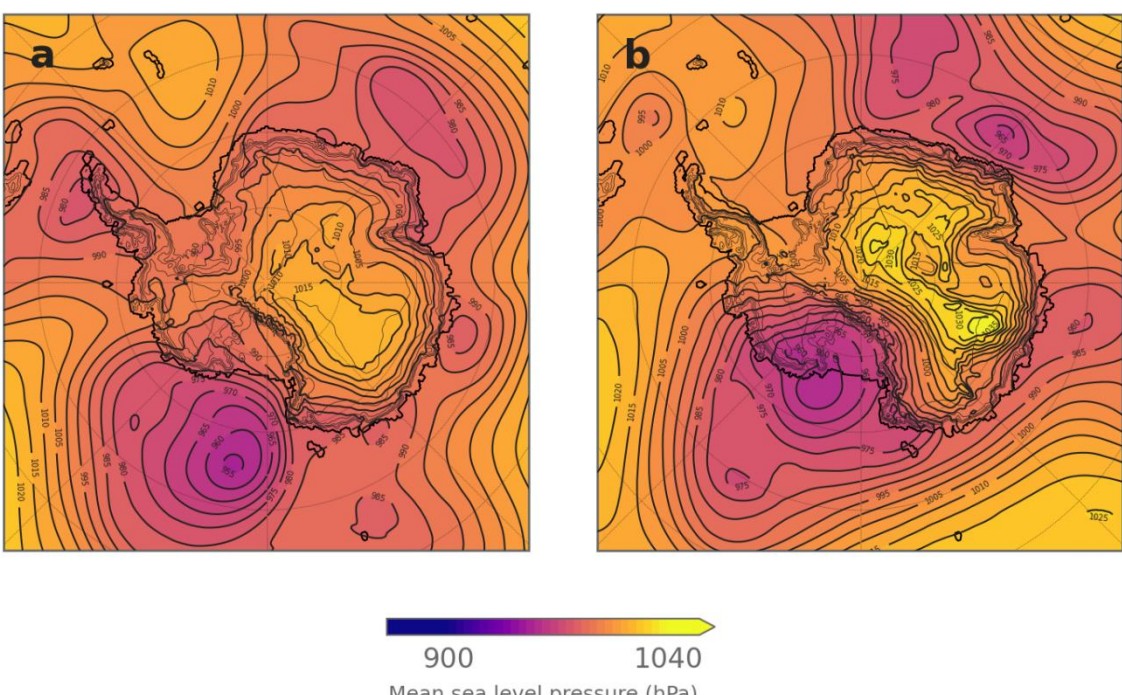

**Figure 2.** Mean sea level pressure at the onset of the summer (a) and winter (b) cases, from the ERA5 reanalysis. Contours show mean sea level pressure in hPa, averaged over the first 24 hours of each case. Surface orography is also indicated by the grey contours.




**Table 1.** Summary statistics for key variables over the Thwaites and Pine Island ice shelves during the summer and winter case studies, as simulated by the three RCMs. Multi-model means of ice shelf-averaged median and maximum values are shown, with the range of RCM values shown in brackets.

| | Air temperature (°C) | | Accumulated snow amount (mm w.e.) | | Accumulated rain amount (mm w.e.) | |
|---|---|---|---|---|---|---|
| *Summer* | Median | Maximum | Median | Maximum | Median | Maximum |
| *Thwaites* | -0.33 | 2.61 | 46.50 | 77.82 | 10.90 | 40.06 |
| | (-0.8 – 0.1) | (2.1 – 3.5) | (35.9– 55.1) | (50.0 – 96.4) | (2.5 – 16.9) | (13.2 –52.2) |
| *Pine Island* | 0.06 | 5.70 | 15.54 | 49.18 | 13.51 | 50.0 |
| | (-0.9 – 0.8) | (4.3 - 6.7) | (13.8-17.1) | (44.4 – 83.3) | (6.2 – 21.9) | (25.0 – 45.2) |
| *Winter* | | | | | | |
| *Thwaites* | -12.12 | -0.04 | 49.90 | 89.4 | 2.58 | 7.77 |
| | (-13.3 to -10.6) | (-0.5 – 0.6) | (35.8-62.7) | (64.5-102.4) | (0.01 - 4.5) | (0.05 –15.8) |
| *Pine Island* | -10.75 | 1.62 | 20.13 | 53.91 | 0.71 | 6.00 |
| | (-11.7 to - 10.2) | (1.3 – 1.9) | (10.1 – 30.8) | (44.4 – 63.9) | (0 – 1.1) | (0.8 – 8.7) |






## 3.2 Precipitation

### 3.3.1 Time series of precipitation

The amount of snow and rain that falls over the PIG and TG ice shelves during the cases examined can tell us about the characteristics of extreme precipitation in the region. Here, we examine SNOWPACK-simulated total accumulation (shown as snow water equivalent, SWE), which comprises total precipitation minus losses via sublimation and runoff. We compare with precipitation amounts from ERA5 and the three RCMs. Figure 3 presents SNOWPACK-simulated SWE at the two stations alongside modelled (RCM + ERA5) ice-shelf average snowfall water equivalent amounts. An equivalent plot, which

includes satellite-derived estimates of total precipitation, is shown in Figure S3. As shown in Figure 3a, summertime SNOWPACK-simulated SWE shows considerable differences between the two stations at Cavity and Channel, despite them being located just 4 km apart. In summer, higher overall SWE totals are simulated by SNOWPACK at Cavity than Channel, and SWE follows a different pattern at the two stations. As shown in Figure 3a, in terms of precipitation totals at Cavity, the first AR (3 February) is indistinguishable from the second (5 February): both bring a considerable amount of precipitation,

driving steady and rapid accumulation between 3-6 February, followed by a pause and then another accumulation event associated with the third AR (8-9 February). This contrasts with the pattern for Channel shown in Figure 3a, which exhibits more modest overall accumulation totals and which increases in three distinct steps associated with each AR, with the first bringing relatively little snowfall in comparison to the second and third ARs in the sequence.

Reasons for the notable differences between SNOWPACK-simulated differences at Cavity and Channel could be localised geographic factors, the spatial characteristics of the ARs, or spatial heterogeneity in accumulation across the TG ice shelf. Firstly, the Channel Camp station is located in the surface expression of a basal melt channel which may experience higher rates of melting, and therefore lower calculated accumulation (Maclennan et al., 2023). Meanwhile, the Cavity Camp station is situated on a flat region of the TG eastern ice shelf. Secondly, the ARs each made landfall on slightly different parts of the

TG ice shelf, with the first falling on the western side of the TG, the second progressing from west to east across TG, and the third hitting the eastern flank of TG (Maclennan et al., 2023). Additionally, the influence of extremely localised and topographically dependent processes such as blowing or drifting snow could have played a role in redistributing snow and therefore impacting accumulation at the two stations.

In summer, in comparison to SNOWPACK simulated accumulation, all RCMs under-estimate snowfall and do not capture the exact timing of the snowfall related to the ARs (Figure 3a). All three do better at Channel, with the MetUM and Polar-WRF simulating approximately the same amount of snowfall and HCLIM exhibiting lower amounts. The multi-model mean ice shelf median cumulative snowfall amount over TG ice shelf is 47 mm w.e. in summer (Table 2), which compares with SWE of 108 mm w.e. and 80 mm w.e. simulated by SNOWPACK at Cavity and Channel, respectively, making the multi-

model mean RCM estimate of median snowfall amount over TG ice shelf approximately two times smaller than the *in situ*



observations suggest. Meanwhile, the median cumulative ERA5-derived estimate of total precipitation over TG ice shelf in the summer case is even lower than the mean RCM estimate, at 28 mm w.e., although still within the range of the RCM mini-ensemble (Figure 3a).

In winter, a similar pattern is observed, with considerable differences between SNOWPACK-simulated snowfall between the Cavity and Channel stations and mismatches in terms of the timing of large snowfall deposits (Figure 3e). However, this time, it is the Channel station that has higher accumulation totals than Cavity. The multi-model mean ice shelf median cumulative snowfall amount over TG ice shelf is 50 mm w.e. in winter, compared with 61 mm w.e. and 116 mm w.e. simulated by SNOWPACK at Cavity and Channel, respectively (Table 2). This means the RCMs compare slightly more favourably 300 against observations in winter than in summer. Wintertime ERA5 values are again at the lower end of the range of RCM estimates over both ice shelves.

However, it must be noted that the snowfall/accumulation values shown in Figure 3 are not exactly comparable because the snow height sensor measurements used to drive SNOWPACK assume all positive height differences are related to snowfall 305 and processes that reduce accumulation, such as rain, melt, runoff, evaporation, wind-driven erosion or sublimation cannot be directly measured. As such, all these processes impact the observed snow height and SNOWPACK assumes that the effects of melt and sublimation are calculated correctly in its computation of the accumulation totals. Although we cannot distinguish their individual impacts using these data, we cannot assume that these processes all have a negligible contribution to the overall accumulation totals during these cases.






**Figure 3**. Time series of median snow (panels a,c,e,g) and rain (panels b,d,f,h) amounts (shown as water equivalent, in mm w.e.) during the summer case study on the Thwaites (a-d) and Pine Island (e-h) ice shelves. Median results from the 1 km MetUM, Polar-WRF and HCLIM domains are shown in red, blue and green, respectively, and ERA5 results are given in yellow. For all models, the shaded regions show the 5$^{th}$ to 95$^{th}$ percentile range of all values simulated across each ice shelf. SNOWPACK-simulated snow amounts derived from observations at the Cavity and Channel stations are shown in panel a) as the dashed and dotted black lines, respectively. Data are only shown until 0900 on 30 June 2020 in panel e due to instrument malfunction after this point. Note the different scales between summer and winter, and between snow and rain.



We also compared the ERA5, RCM and SNOWPACK-simulated totals with the GPM IMERG satellite product (see text S2

for further details). However, due to the product's limitations over snow and ice-covered surfaces, as well as its reliance on reanalysis data, we chose not to include it here. A figure comparing the satellite estimates to the model outputs shown in Figure 3 is given in Figure S3. It shows that during both cases, the satellite-derived estimates of total precipitation are extremely close to ERA5 (reaching median values of 28 mm w.e and 19 mm w.e. over the TG and PIG ice shelves in summer), and considerably lower than the RCM and SNOWPACK estimates. The similarity to ERA5 is expected because the

precipitation product is partly generated using reanalysis data that has assimilated the same observations as ERA5, and ERA5 also assimilates satellite data. That the satellite-derived estimates of precipitation are lower than the multi-model mean of the RCM estimates is consistent with the lower resolution of the satellite product ($0.1° \times 0.1°$ or approximately 12 km) and may also reflect the difficulty of satellite sensors in measuring precipitation over cold snow and ice surfaces. This underlines the challenges associated with evaluating precipitation extremes in Antarctica and reinforces the merit of high-resolution RCMs.


Neither ERA5 nor the satellite measurements shown in Figure S3 appear capable of capturing the spatial variability over the TG ice shelf that is implied by the large differences between the Channel and Cavity observations shown in Figure 3a and Figure 3e. SNOWPACK-simulated accumulation at Channel in Figure 3a and at both stations in Figure 3e is mostly within the 5-95th percentile range of the RCMs and ERA5, but values at Cavity in summer and at Channel at the end of the winter

case are well outside the modelled range, indicating that despite their high resolution, the RCMs still under-estimate snowfall amounts compared to *in situ* observations. Although the RCM 5th-95th percentile range demonstrates that the RCMs can capture the magnitude of spatial heterogeneity implied by the differences between Cavity and Channel, the time series at grid points corresponding to each station's location (not shown) are extremely similar, suggesting that there are still unresolved spatial variations below the 1 km grid scale of the RCMs, or processes that are incorrectly represented. This may be partly

because there are important processes that are unresolved in the RCMs, such as blowing or drifting snow, or because of deficiencies in important parameterisations such as cloud microphysics. Furthermore, small-scale variations in orography can cause blowing or drifting snow to accumulate in some locations preferentially, while the scale of localised and rapidly evolving depressions or topographic features is below the grid scale of the models and may not be included in the datasets used to produce static boundary conditions such as surface altitude. The spatial variability of snow and rainfall over the

domain is explored in greater detail in section 3.3.3 below.

However, it must be noted that the difference between RCM and SNOWPACK snowfall totals, particularly in the summer case, is of approximately the same order of magnitude as the amount of rain simulated by the RCMs. This indicates that phase partitioning could be one of the main causes of the discrepancy between RCM-simulated and SNOWPACK-simulated

snowfall. This may be because SNOWPACK uses a fairly simplistic temperature threshold to diagnose rain, based on when ERA5 2 m temperature reaches 1.2°C. As shown in Table 2, the RCMs exhibit much warmer conditions over the TG ice shelf than ERA5, suggesting that rain would have been diagnosed much less frequently based on ERA5 temperatures than the



RCM temperatures. Therefore it is possible that SNOWPACK underestimates rainfall: a conclusion also supported by the fact that all three RCMs simulate summertime rain and that two of the three simulate rain in winter. Note that when

SNOWPACK diagnoses rainfall, this is additional to the accumulation derived from snow depth measurements, and calculated accumulation is unchanged.

### 3.3.2 Precipitation phase

We next examine the phase partitioning of precipitation in the RCMs over the TG and PIG ice shelves during both cases. During both cases, considerably more snow falls over the TG ice shelf than the PIG ice shelf in all models, with median

snowfall totals on average approximately 2-3 times higher over the TG ice shelf than the PIG ice shelf (Table 1). Meanwhile, more rain falls over the PIG ice shelf in the summer case than the TG ice shelf, whereas in the winter case the TG ice shelf is rainier (Table 1).

Table 2 shows that all three RCMs simulate rainfall in summer, with 2.5 – 16.9 mm w.e. rain falling over the TG ice shelf and

3.1 – 12.4 mm w.e. falling over the PIG ice shelf. Further, two out of three RCMs simulate rain in winter, with 0.01 – 4.5 mm w.e. falling over the TG ice shelf and up to 1.1 mm w.e. over the PIG ice shelf in winter. Polar-WRF and the MetUM both simulate upwards of 1 mm w.e. in winter, while HCLIM simulates negligible amounts. The RCMs simulate the most rain at the end of the summer case, particularly on the 7/8[th] February. However, the raw output from the snow height sensors on TG ice shelf (Figure 4) indicates a period of accumulation during this timeframe (accumulating ~11 mm), which appears incompatible

with considerable rainfall. Typically, rain reduces the height of the snowpack by compacting snow (Marshall et al., 1999), which means that significant rainfall is unlikely on the 8[th] February. However, there may be several explanations for this apparent inconsistency between the observations and model results.

The time series shown in Figure 3a and b indicate that there were two periods of intense precipitation over TG ice shelf at the

end of the summer case: first from 1800 on 7 February until 0000 8 February, and then from around 0600 until 1800 on 8 February. During that first 6-hour window, the RCMs simulate minimal snowfall and relatively significant rainfall amounts: 6-7 mm w.e. in Polar-WRF and the MetUM, compared with 2-3 mm w.e. snowfall (Figure 3a, b). However, during the second 12-hour window, both snow and rain are simulated, with the MetUM and Polar-WRF simulating around 15 mm w.e. of snowfall, compared with 5-6 mm w.e. rainfall. Given that the near-surface atmospheric conditions during this case were near the melting

point (see Figures 6 and S1), with warm air overlying the snowpack, it is likely that precipitation was mixed phase. As a result, it is plausible that snow, rain and wet solid precipitation such as wet snow or sleet all fell during this time. However, the RCMs used in this study have a relatively simplistic categorisation of hydrometeors: the MetUM represents five while Polar-WRF represents four (HCLIM does not simulate considerable rain or snow during this period). Therefore, wet solid precipitation may have been classified as 'rain' in the RCMs, increasing the amount of rainfall simulated. As long as the amount of snow falling

was comparatively greater than the amount of liquid precipitation, the height of the snowpack could still have increased, but the



amount of rain by mass would have been small. However, without a ground-truthing of rainfall from *in situ* measurements, it is difficult to say for certain whether rain indeed fell during these days.



**Figure 4**. Smoothed measured snow height and SNOWPACK-simulated cumulative rainfall amounts in the summer (a) and winter (b) cases. Plots show measured snow height change (in m) in black and rainfall amounts (in mm) in blue. Both the Cavity and Channel stations are shown.





Considering the entire summer case, it seems plausible that a small amount of rain did fall. Indeed, this is supported by the
SNOWPACK output, which indicates that 1.5 and 1.6 mm fell at Channel and Cavity, respectively, during the summer case,
including 0.9 mm w.e. at 0000-0600 UTC on the 8<sup>th</sup> February at both stations (Figure 4). The RCMs may have simulated rainfall
at the wrong time or misclassified wet solid precipitation as rain. Further, as discussed above, the simple, ERA5-derived
temperature threshold used to diagnose rain could cause SNOWPACK to underestimate rainfall compared to the RCMs.
Additionally, any rain that fell at sub-zero temperatures (i.e. supercooled rain) would not be captured using this 1.2°C threshold.
The same explanations probably apply to simulated winter rainfall too, although winter rainfall totals are much smaller in
comparison to snowfall and compared to the summer case. The caveat is that there are no *in situ* observations of rainfall to
provide a ground truthing of this result. However, the fact that rain is simulated by all three RCMs strengthens the notion that
rain could indeed be falling during these extreme events and emphasises the need for *in situ* measurements of liquid as well as
solid precipitation in West Antarctica.

**3.3.3 Spatial variability of precipitation**

The spatial variability of accumulated precipitation over the entire ASE region is also important. Figure 5 shows maps of
accumulated snow and rain in ERA5 and the three RCMs during the two cases. In summer, the largest quantities of snow are
simulated in the west of the domain (bottom right of all panels in Figure 5a-d), and along the windward slopes of the steepest
terrain, for example the sloping topography on either side of the Abbot ice shelf, and the upwind slopes above the PIG and
TG ice shelves. The total magnitude of accumulated snowfall is comparable between all three RCMs and ERA5, with the
MetUM and Polar-WRF standing out as the snowier models. A similar pattern is clear during winter (Figure S4). Figure 5e-
h shows that rain also falls in the vicinity of steep terrain, but unlike snow, rain appears more confined to lower elevations.

Figure 3b, d and Figure 5 both show that all the RCMs simulate rainfall over the TG ice shelf and PIG ice shelf during
summer, while ERA5 simulates rainfall over the PIG ice shelf only. During the summer case, an RCM median of 11 mm w.e.
(2.5 - 17 mm w.e.) rain falls over the TG ice shelf, while 13.5 mm w.e. (6 - 22 mm w.e.) falls over the PIG ice shelf (Table
1). Polar-WRF has relatively higher median rainfall totals over the PIG ice shelf than the MetUM and HCLIM (Figure 3d),
but if we consider the spatial distribution of accumulated rainfall shown in Figure 5, the MetUM and Polar-WRF are extremely
similar in terms of cumulative rainfall amounts. However, rain is simulated in different locations between the two models,
with Polar-WRF simulating higher quantities of rainfall over the steepest terrain of Thurston Island on the northwestern
boundary of the Abbot Ice Shelf and more extensive rainfall over the PIG ice shelf. Meanwhile, the MetUM simulates pockets
of higher rainfall near the windward side of steep terrain, including at the edge of the PIG and TG ice shelves where the
terrain begins to slope upwards more strongly. HCLIM also simulates pockets of rainfall clustered near steep topography,
particularly on Thurston Island and over the PIG ice shelf, but comparatively lower accumulated totals than in the other two



RCMs. ERA5 accumulated rainfall totals are highest along the steep periphery of Thurston Island and the southwestern edge of the PIG ice shelf, but overall much lower than all three RCMs.

The same pattern is observed in winter, with the MetUM and Polar-WRF simulating larger quantities of rainfall than ERA5 or HCLIM, but in contrast to the summer case, the MetUM stands out as the rainier model, simulating a band of rain of up to 430 30 mm w.e. along the spine of the steep topography of Thurston Island (Figure S4). More rain is simulated by all three RCMs during the summer case, because temperatures are higher and more precipitation falls in general. However, two out of three simulate modest quantities of rain during the winter case, which may indicate that in the most extreme AR conditions, liquid precipitation can still occur, even in winter.

The following sections will explore the origins and physical processes leading to the production of liquid precipitation during the extreme events simulated.

**Table 2.** Median air temperature in degrees celsius and median accumulated total snow and rainfall amounts in mm water
equivalent (mm w.e.) for each model and ERA5 over the Thwaites and Pine Island Glacier ice shelves in the summer and winter cases.

| | Thwaites | | | PIG | | |
|---|---|---|---|---|---|---|
| *Summer* | Air temperature (°C) | Snow (mm w.e.) | Rain (mm w.e.) | Air temperature (°C) | Snow (mm w.e.) | Rain (mm w.e.) |
| *MetUM* | -0.3 | 55.1 | 16.9 | 0.3 | 15.8 | 6.2 |
| *WRF* | 0.1 | 48.5 | 13.4 | 0.8 | 13.8 | 21.9 |
| *HCLIM* | -0.8 | 35.9 | 2.5 | -0.9 | 17.1 | 12.4 |
| *ERA5* | -10.7 | 27.9 | <0.1 | -11.8 | 18.9 | <0.1 |
| *Winter* | | | | | | |
| *MetUM* | -12.4 | 51.2 | 4.5 | -11.8 | 10.1 | 1.1 |
| *WRF* | -13.3 | 62.7 | 3.2 | -10.2 | 19.5 | 1.0 |
| *HCLIM* | -10.6 | 35.8 | <0.1 | -10.3 | 30.8 | <0.1 |
| *ERA5* | -21.1 | 29.8 | 0.0 | -21.6 | 19.9 | 0.0 |





**Figure 5.** Accumulated total snow (panels a-d) and rain (panels e-h) during the summer case, as simulated by ERA5, the MetUM, Polar-WRF and HCLIM at 1km resolution. An equivalent figure for winter is shown in Figure S4.



### 3.4 Origins of liquid precipitation in ARs

We have established that liquid precipitation is likely falling during the cases examined over the TG and PIG ice shelves. Next, we explore the origins of this rainfall. To avoid repetition, the following analysis will use data from the 1 km MetUM simulation only, which exhibits some of the highest rainfall totals.

Figure 6 shows the relationship between maximum air temperatures and accumulated rainfall totals during the winter and

summer cases in the MetUM. Maximum temperatures reach well above freezing at the lowest elevations in summer, especially over the PIG ice shelf, where maximum temperatures of ~5°C are simulated across much of the ice shelf nearest the surface (Figure 6a). However, maximum temperatures above zero are also simulated during winter over the Abbot, Cosgrove and PIG ice shelves (Figure 6d). These above-freezing temperatures are simulated over regions in the immediate lee of steep terrain, suggesting that a foehn effect could be altering the meteorology associated with the ARs, as also shown by Francis et al. (2023)

over PIG, by Gehring et al. (2022) over Davis, by Wille et al. (2022) over the Antarctic Peninsula and Lu et al. (2023) over the South Orkney Islands. A foehn effect could be expected to produce a warm pool near the surface in the lee of steep terrain when air flow is across this topography.

Panels b and c of Figure 6 show that there is indeed a warm pool of air at the surface over the TG and PIG ice shelves, and

that the region of above-freezing summer maximum mean air temperatures along the transects through TG and PIG reaches up to above 1 km in altitude, and up to nearly 2 km altitude over PIG. These higher temperatures aloft may be partly associated with a foehn effect, and/or by the air mass being orographically lifted, resulting in cooling, condensation and latent heating of the surrounding atmosphere. In winter, regions of above-zero maximum air temperatures are much more spatially confined and are concentrated in small pockets over the PIG ice shelf at altitudes below 750 m (Figure 6e), while temperatures never

reach the freezing point along the TG transect (Figure 6f).

Although near-surface temperatures over the PIG and TG ice shelves reach maximum values near or above freezing during both cases (Table 1, Figure 6), median temperatures are colder, remaining below freezing during the winter case and staying very close to zero in the summer case (Table 1). Further, most of the liquid precipitation falls at times and altitudes when

ambient temperatures are not above freezing, suggesting that temperatures above 0°C are not always required to produce rainfall.

The dashed lines in Figure 6 indicate the maximum height of the 0°C air temperature threshold and can be considered a maximum 'melt line' height. Below this threshold, snow and ice might be expected to melt and fall as rain if the melt layer

that solid particles fall through is thick enough. However, in the summer case, the highest accumulated rainfall totals are not always concentrated below the melt line, especially in the case of the TG transect. Moreover, in the winter case the





region of above-zero temperatures is extremely spatially constrained along the PIG transect and completely absent along the TG transect. Despite this, accumulated rainfall totals in excess of 10 mm w.e. are simulated along the transects through both ice shelves, resulting in summer surface rainfall totals of 16.9 mm w.e. and 6.2 mm w.e. over the TG and PIG ice
shelves, respectively, and 4.5 mm w.e. and 1.1 mm w.e., respectively, in winter (Table 2). This means that supercooled liquid precipitation is falling, even fairly high up in the atmosphere.

Over both ice shelves and in summer especially, accumulated rainfall totals are higher just above the surface than at the surface, indicating that some rain is being lost to evaporation, riming or freezing as it descends through the warm near-
surface layer, or that rain droplets are being lofted upwards by updrafts as air is forced to flow up the steep terrain of the ice stream, as described in Silber et al. (2019). Meanwhile in winter, most of the rainfall that reaches the surface is spatially constrained to further along the TG transect, upstream of the grounding zone where the AR intersects with steep topography. Over the PIG ice shelf winter rainfall is more minimal, despite temperatures being warmer than over the TG ice shelf, as indicated in Tables 1 & 2.


As shown in Figure 6, patterns of accumulated rainfall are relatively consistent between locations and seasons. In both seasons, rain is present at altitudes of up to ~2500 m, although in summer, the height of the 5 mm w.e. accumulated rainfall contour is marginally higher, reaching up to around 3000 m a.s.l. (Figure 6b, c). This is despite the patterns of maximum air temperature being quite different between the transects. In both seasons, there are also pockets of high accumulated
snowfall totals above the surface over the ice shelves (Figure S5), indicating that in some locations snow is sublimating and/or melting into rain as it descends through the warm layer described above. This same effect has been observed during combined AR / foehn conditions at Davis station in East Antarctica by Gehring et al. (2022).





**Figure 6**. Characteristics of maximum air temperature and accumulated rainfall during the summer case (top row) and the winter case (bottom row). Panels a and d show maximum 1.5 m air temperature during each case, with the locations of the PIG and Thwaites transects indicated on the map. Panels b and e show transects of maximum temperature (colours) and accumulated total rainfall in mm w.e. (contours) across the Thwaites transect and panels c and f show the same across the PIG transect. In all transect panels, the dashed line indicates the maximum height of the 0°C air temperature threshold, and in all panels, red colours indicate maximum air temperatures above 0°C, while blue colours indicate temperatures colder than this threshold.

Liquid precipitation falls from liquid-bearing mixed phase cloud at temperatures as cold as -11°C and -8°C in winter and
summer, respectively (not shown), suggesting that the formation mechanisms are supercooled warm-rain processes. Cloud
droplets reach a large enough size to descend through the atmosphere, collecting other droplets on their way and producing
supercooled rain through warm-rain processes like collision-coalescence and accretion. Supercooled drizzle usually forms
when there are insufficient ice nuclei to allow the atmospherically available water vapour to nucleate and grow into solid
precipitation via ice processes. This is also consistent with a pristine region such as the West Antarctic, and the





norththeasterly wind direction which has advected air from across Ellsworth Land where it is unlikely to have picked up ice nucleating particles. These results also suggest that rainfall is being generated via the seeder-feeder mechanism, which increases orographic rainfall rates, and can often result in the formation of supercooled drizzle in mixed-phase clouds (Fernandez-Gonzalez et al., 2015; Ramelli et al., 2021). The process involves ice or liquid descending from upper (seeder)

levels and enhancing the rate of accretion in lower (feeder) levels, thereby increasing surface rainfall amounts (Lean & Clark, 2003). It frequently occurs in regions with significant updrafts, such as when air is forced up and over steep terrain (Ramelli et al., 2021). Finlon et al. (2020) show that the 0°C isotherm separates the seeder and feeder levels in such cases, for example in an AR over the Southern Ocean, which is consistent with the results shown here.

In both cases, the region of accumulated rainfall reaches several hundred kilometres inland. Maximum air temperatures are higher over the PIG transect as shown in Figure 6a, but in summer, higher accumulated rainfall totals are simulated over the TG ice shelf and upstream (Figure 6b). This suggests that air temperatures are not the only control on rainfall totals, and that factors like large-scale cloud and synoptic patterns, as well as orography, likely influence the phase and quantity of precipitation that falls.


This is also supported by Figure 7, which shows the ratio of rain to snow over the TG and PIG transects in summer and hints at different formation mechanisms of precipitation. Hatched regions in Figure 7a show that there are several regions where the amount of summertime rainfall exceeds the amount of snowfall: most of these regions are offshore, with some also immediately above the steepest slopes in the lee of airflow over the terrain, for example over the PIG and Abbot ice

shelves. Along the TG and PIG transects shown in panels b and c, summer precipitation is largely mixed phase and accumulated totals are dominated by snowfall, but rain/snow ratios of above 0.5 (indicating that at least 33% of precipitation is falling as rain) highlight regions where rainfall totals are relatively high and liquid precipitation processes may be comparatively more important. Figure 7b shows that along the TG transect, the highest rain/snow ratio is found above the zero-degree line at higher elevations around 150 km inland above the glacier slope. However at the surface, the highest

rain/snow ratios are simulated at the foot of the steep terrain, as indicated by Figure 7d.

Along the PIG transect, much more snow falls relative to rain (Figure 7c), except at the lowest elevations and directly above the steepest orographic gradient (Figure 7e), where the effect of foehn-driven warming is highest (Figure 6a, b). Over the PIG transect, the highest rain/snow ratios are found in the warmest regions, indicating that some of the rain has formed by

the melting of solid precipitation as it descends through the warm near-surface layer (Figure 6c). Some of the precipitation likely also sublimates and evaporates as it descends through this warm layer, which is illustrated by the fact that the highest rainfall fluxes are simulated in pockets above the surface in both winter and summer.



The importance of super-cooled rain formation processes is especially evident in the winter case, where maximum air
temperatures above 0°C are simulated only in extremely confined locations over the PIG ice shelf (Figure e). Supercooled
liquid precipitation of up to more than 10 mm w.e. occurs over both ice shelves, especially at higher altitudes. This compares
with maximum cumulative totals of ~20 mm w.e. in summer, i.e. double the magnitude of winter.

The presence of supercooled rain over these two important glaciers/ice shelves is difficult to verify in the absence of *in situ*
rainfall measurements. However, the fact that rain is simulated by all three RCMs, ERA5 and SNOWPACK in summer and
two out of three RCMs in winter lends strength to the idea that liquid precipitation indeed fell during these particular extreme
events. Moreover, liquid precipitation has been shown to occur at very low temperatures in Antarctica. For example, Silber
et al. (2019) demonstrated that drizzle can occur at temperatures down to -25°C near McMurdo. They showed that in the
presence of sufficient condensate, liquid precipitation can fall from stratiform clouds like those that were present during the
summer case. In this instance, the source of condensate was from the three Ars that made landfall, as well as the orographic
lifting and condensation of moisture in the air as it was forced over the steep topography. In the pristine Antarctic
atmosphere there are also fewer aerosols that can act as ice nucleating particles, thus preventing the rapid glaciation of the
cloud.



**Figure 7.** Ratio of accumulated rain to snow during the summer case, expressed as rain / snow. Panel a) shows the mean ratio of accumulated rain / snow at the first model level, while panels b and c show transects of accumulated rain / snow along the transects through Thwaites Glacier and Pine Island Glacier indicated in panel a. Regions on the map in panel a where the ratio is above 1:1 (indicating that total rainfall is equivalent to or higher than total snowfall) are hatched. The dashed contour on panels b and c shows the maximum height of the 0°C air temperature threshold as in Figure 7, while the white contour outlines the region where the ratio of rain to snow is 1:2 or higher. Note the difference in colour scales used between panel a and panels b and c. Panels d and e show the rain:snow ratio at the surface along the Thwaites glacier (panel d) and Pine Island Glacier (panel e) transects, with the height of elevation indicated in black.



While rainfall does not occur frequently over the ASE in the current climate and is exclusively associated with extreme events like the ARs described here, these conditions may occur more often in future. Antarctic total precipitation amounts will increase with warming (by 5.5% per degree Celsius on average) and the number of extreme precipitation events and ARs over the ASE will probably continue to rise with global temperatures (Nicola et al., 2023). There is also some evidence that climate change may cause ARs to shift poleward (Ma et al., 2020). Although ARs are unlikely themselves to destabilise ice shelves like the

TG ice shelf, they affect accumulation and so could change the timing of future mass loss from West Antarctic glaciers (Maclennan et al., 2023). For example, Donat-Magnin et al. (2020) use RCMs to show that ablation will increase over ASE ice shelves such as the PIG and TG ice shelves, primarily driven by increased melt and runoff rates, but that the ratio of melting to snowfall is still low compared to other Antarctic ice shelves like the Ross or Filchner-Ronne. This is corroborated by Gilbert & Kittel (2021), which shows that even under a high-emissions scenario, the risk of surface-driven West Antarctic ice shelf

collapse is low until the end of the century because the amount of runoff is greatly outweighed by increased snowfall.

However, climate change will also increase the proportion of precipitation falling as rain, as has been shown for the Arctic (McCrystall et al., 2021). For instance, Donat-Magnin et al. (2021) show that rain occurs on 0.5 and 1 day per year over the TG and PIG, respectively, in the current climate but that this may increase to 4.2 and 5.8 per year by 2100. These changes to

total precipitation and the phase of precipitation are extremely important for determining ice sheet and ice shelf SMB. For example, rain-on-snow events can cause the surface to darken, reducing surface albedo and enhancing melt via the melt-albedo feedback (Box et al., 2022; Stroeve et al., 2022). As the climate warms, the future of Antarctic precipitation phase will remain an extremely pertinent subject of research.

## 3.5 Impact of resolution on precipitation

As noted in section 1, model resolution can strongly determine the amount of precipitation simulated, especially in regions with steep topography and where precipitation is dominated by orographic processes, such as those we have shown to be occurring here.

As shown in Figure 3 and Table 2, the RCMs simulate higher accumulated snow and rainfall totals than ERA5, which is likely

related to resolution. At ~31 km, the resolution of ERA5 is considerably coarser than the resolution of the 1 km innermost RCM nests. One of the variables for which this is most apparent is air temperature. As shown in Table 2, all three RCMs indicate much warmer median air temperatures over the TG and PIG ice shelves than ERA5, suggesting the lower resolution of ERA5 is not sufficient to capture the topographic modification of the AR. The absolute maximum temperatures reached are also reduced, and this is also clear from the fact that median air temperatures over the PIG and TG ice shelves are approximately

10°C lower in ERA5 than the RCMs in both cases (Table 2). The agreement between the RCMs, all of which are forced by ERA5, suggests that the representation of complex topography is the cause of this difference, rather than internal model differences. This indicates that high resolution is necessary to capture the complex interactions between large-scale, mesoscale



and local meteorological processes which are important during such conditions in this region, and that RCMs are required to reproduce the factors governing the drivers and impacts of AR events in regions with complex topography.


However, there is a trade-off when conducting simulations between resolution and computational efficiency, so it is worth investigating what kind of resolution is sufficient to capture the dynamics of extreme precipitation in this region. To explore this, Figure 8 shows accumulated snow and rainfall totals during the summer case across the three different resolution nests of the MetUM: 1 km, 3 km and 12 km grid spacing.


There are few differences in accumulated snowfall amounts between the various resolution domains of the MetUM. Given that snowfall dominates over rainfall, this suggests that the model can capture the dominant drivers of extreme precipitation, even at the relatively coarser resolution of 12 km. However, there is still some advantage to using higher resolution to capture finer-scale spatial variability when concentrating on smaller regions such as a single ice shelf. Furthermore, while
total accumulated snowfall in the 12 km simulation is comparable to 1 km totals, the accumulated rainfall totals are lower in the 12 km simulation compared to the 3 km and 1 km simulations (Figure 8), with rain simulated only over the steepest terrain, such as Thurston Island and King Peninsula. Because considerably less rain falls than snow, this has a fairly minimal effect on total accumulated precipitation. Nonetheless, this suggests that sub-grid scale processes may not be parameterised adequately at coarser resolutions, for example the sub-grid phase partitioning of cloud, or the microphysical processes
controlling the generation of supercooled drizzle. These are known problems with the MetUM (Gilbert et al., 2020; Bodas-Salcedo et al., 2016; Abel et al., 2017;) and other RCMs (Hyder et al., 2018; Vergara-Temprado et al., 2018), especially in Antarctica and the Southern Ocean. This suggests that while coarser resolution configurations of RCMs can capture the large-scale characteristics of precipitation extremes, finer grid scales are required to fully capture the complex interactions producing liquid precipitation in this region.




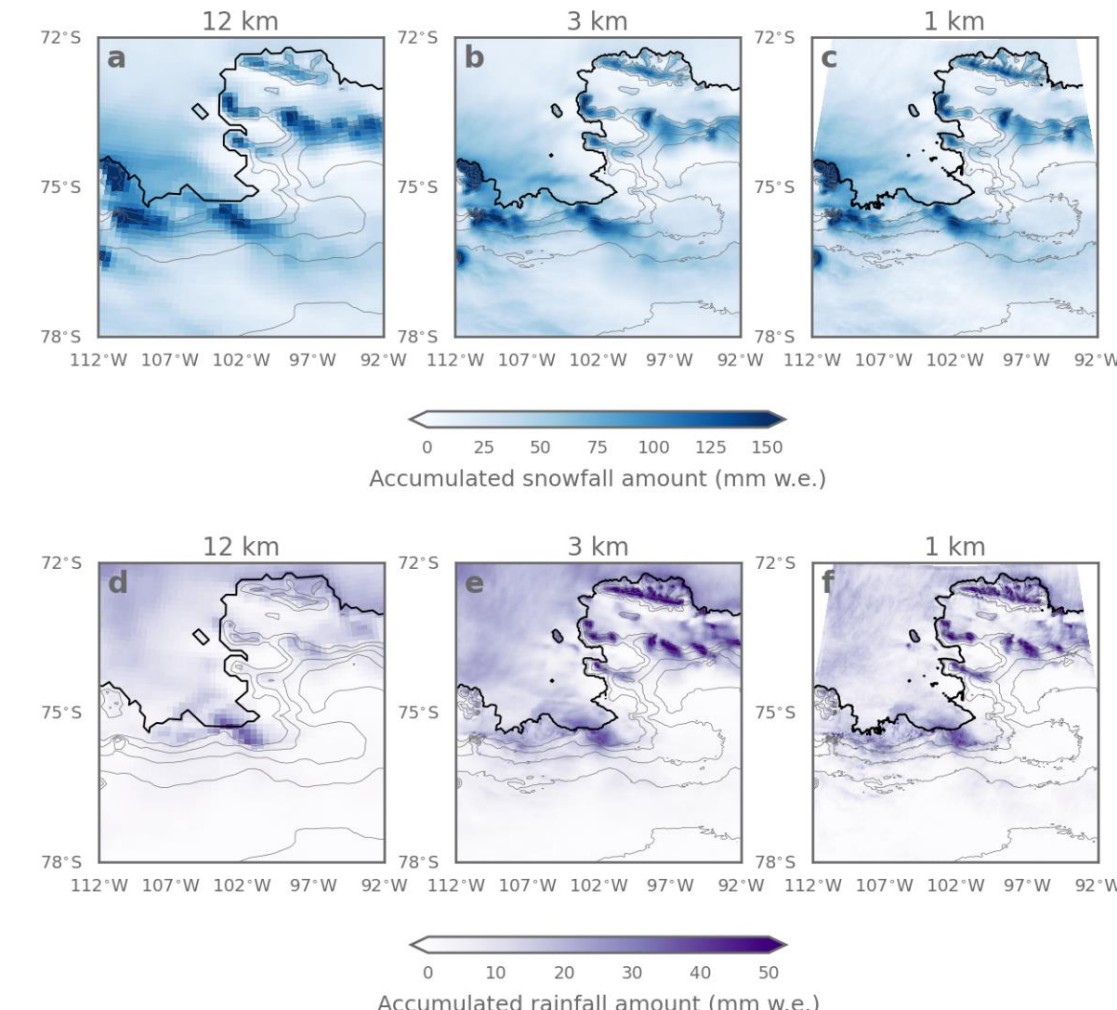

**Figure 8.** Accumulated snowfall (a-c) and rainfall (d-f) during the summer case, as simulated by the MetUM at 12 km, 3 km and 1 km horizontal grid spacing.




## 4 Conclusions

We have used a mini-ensemble of state-of-the-art RCMs to explore the characteristics of extreme precipitation events in West Antarctica. Precipitation totals are especially high when the AR interacts with complex topography, generating snow and rain as a result of orographic uplift and allowing warm conditions favourable to liquid precipitation to persist via latent heating and foehn conditions near the surface.

We have shown that estimates of precipitation produced by the three RCMs are consistent with observations and with each other. For example, they simulate snowfall totals that are of comparable magnitude to *in situ* measurements of snow accumulation. However, all RCMs underestimate the amount of snowfall at both sites and do not capture the observed differences between stations, suggesting that there are highly localised, unresolved processes influencing snowfall.

The broad patterns of rain and snowfall are captured in ERA5 and all RCMs. Considering ice shelf-averaged totals as well as the spatial distribution of precipitation, Polar-WRF and the MetUM simulate extremely similar cumulative summer snow and rainfall amounts, with HCLIM and ERA5 producing relatively less over the ice shelves of interest. However, during winter, the MetUM simulates more rain over the TG and PIG ice shelves, and much higher rainfall totals over the steepest terrain in the domain. The simulated amount of rain falling in both cases is non-trivial, especially during the summer case
when up to 30 mm w.e. falls over some parts of the domain. However, two of the RCMs indicate that small amounts of rain may be falling even during the winter case, when temperatures are almost entirely below freezing. The lack of *in situ* rainfall observations means this is impossible to verify, but the occurrence of rain in all three RCMs, ERA5 and SNOWPACK in summer, and in two out of three of the RCMs in winter supports the idea that rain is indeed falling over this region.

The RCMs compare more favourably against observations than ERA5 reanalysis, which is unable to capture the dynamics of these events seen in the RCM output. ERA5 has been used extensively to assess the impacts of ARs in Antarctica, but we have shown that it has insufficient resolution to capture the interaction between ARs and steep topography. This is a limitation considering that the most extreme and high-impact events often result from the combination of ARs with topographically induced phenomena like foehn events and may suggest that impacts have been previously underestimated.
We further evaluated the differences between MetUM simulations at varying resolution. The MetUM exhibits a convergence across resolution with respect to total precipitation, indicating that the coarser resolution 12 km domain is sufficient to capture its large-scale dynamics and drivers. However, the 12 km MetUM configuration does not represent the phase of precipitation as accurately as higher resolutions, suggesting that km-scale resolution is required for an accurate assessment of precipitation phase. Furthermore, for very detailed studies of the microphysical processes contributing to
precipitation, finer resolutions – in this case 1 km – offer considerable benefits over configurations at ~10 km scale. This

work therefore emphasises the importance of RCMs for examining the dynamics, origins and development of ARs in regions with complex topography.

Further analysis of the MetUM simulations shows that rain is formed via different processes. In summer in the presence of
a near-surface warm pool, some rain forms when solid precipitation descends through the warm near-surface layer and melts. However, in colder conditions, such as over the TG transect in summer and over both locations in winter, super-cooled warm-rain processes act to form drizzle from liquid-bearing stratiform cloud in sub-zero temperatures. It is beyond the scope of this study to explore either the cloud microphysical processes that produced rain in these cases or the specific impacts of rainfall on the stability of the TG and PIG ice shelves, but both would be fruitful avenues of further research.
These findings emphasise the importance of using high-resolution modelling to fully capture the complex dynamics and processes occurring during these extreme events, as well as the need for further *in situ* precipitation measurements in this region, including of liquid precipitation. As climate change continues, rainfall is likely to become an increasingly regular occurrence and further work must be done to understand its implications for melt-albedo feedback processes, latent heat release, and glacier and ice shelf surface mass balance in this critical region.

**Data availability**

ERA5 data are available via the Copernicus Climate Data Store (Hersbach et al., 2023). RCM data are archived at 10.5281/zenodo.12697647.AMIGOS AWS and snow sensor height data are available at https://www.usap-dc.org/view/dataset/601549 while the SNOWPACK simulations are available at https://doi.org/10.5281/zenodo.7320237. The NASA GPM IMERG product used (in the supplement) is the IMERGHHR Final Precipitation L3 Half Hourly 0.1 Degree x
0.1 Degree V07 product (Huffman et al., 2023) and is available at https://doi.org/10.5067/GPM/IMERG/3B-HH/07.

**Competing Interests**

The authors declare no competing interests.

**Acknowledgements**

EG, JAT, DP, KV and AO acknowledge support from the PolarRES project, which has received funding from the European Union's Horizon 2020 research and innovation programme call H2020-LC-CLA-2018-2019-2020 under grant agreement number 101003590. EG also acknowledges the use of the ARCHER2 UK National Supercomputing Service (https://www.archer2.ac.uk). MM acknowledges support from NASA FINESST grant 80NSSC21K1610.



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
