# Peer review of "Extreme precipitation associated with atmospheric rivers over West Antarctic ice shelves: insights from kilometre-scale regional climate modelling"

_EGUsphere, 2024_

## Referee Comment (RC1)

General comments

This study takes an in-depth look at how kilometer scale regional climate models (RCMs) resolve the interactions between atmospheric rivers (ARs) and complex topography in a couple case studies centered around the Amundsen Sea Embayment. Their most consequential results indicate that the boost in resolution in these three RCMS (MetUM, Polar-WRF, and HCLIM) leads to the resolution of non-trivial rainfall quantities over the Thwaites and Pine Island ice shelves, especially during the summer AR event. The ERA5, normally considered the gold standard for Antarctic precipitation studies, is unable to depict the rainfall quantities from these AR events. The final experiment of using the MetUM shows that even model resolutions considering high (12 km) are too coarse to simulate topographically influenced rainfall on the Antarctic coastline.

The authors make a great effort using SNOWPACK to calibrate the ground observations for a better comparison with the RCMs and ERA5 instead of just comparing the models directly with the station observations. Despite the consideration, there remains a great deal of uncertainty in validating the performance of the RCMs and ERA5 at this high resolution where the authors meticulously highlight the potential shortcomings of studying this remote region. Other efforts like incorporating IMERG rainfall data and the discussing its shortcomings emphasize the level of detail that went into this study. From my perspective, all the most relevant research was cited and is usefully employed to place the results in context with the existing literature. I believe this study conveys the potential unrealized threat of heavy rainfall on Antarctic coastlines and the need for advanced modeling and observations to further realize this problem. I've made specific and technical comments below thar hopefully help the authors clarify a few details, but I believe this manuscript will be an important contribution to the field after some minor revisions.

-Jonathan Wille

Specific comments

Line 23-25: Can you specify that you used the MetUM for the resolution experiment here?

Line 33-34: Can you elaborate on this mass loss from ice shelves? Like are the ice shelves themselves shrinking in mass or is something else happening? Also please include a reference for Smith et al. (2020)

Line 39: The wording is a bit confusing here. If accumulation is not compensating for SMB losses on the ice shelves, then perhaps instead of saying "constrain", try "… but snowfall still

modulates (or controls) the timing and characteristics of ice shelf collapse, or recession and thinning"

Line 70-80: This rainfall paragraph is well-written but it could be beneficial to cite (Vignon et al., 2021) which characterized rainfall patterns around Antarctica (i.e. it happens more often than you'd expect) and uses GCMs to show increased rainfall in the future.

Line 158: Can you state how deep convection is treated in the HCLIM? This is described for the other models so it would be good to know here as well.

Section 2.3: Since this becomes relevant later in the results, can you describe the topography surrounding the stations so that the reader can get a feeling for how orographic lifting might behave here? Also it might be important to state any potential snow measurement issues at these stations like blowing snow?

Section 2.4: Is this method of diagnosing snowfall using SNOWPACK sensitive to blowing snow related errors or would it actually reduce the impacts of blowing snow?

Section 3.1: It would be helpful if you gave more details on the temporal evolution of both case studies. Like providing dates for the AR landfalls. Also at the risk of self-advertising, it could also be helpful to state that these events were detected AR events from the Wille et al. (2021) detection algorithm (I already verified this).

Section S2.1: It was instructive to see IMERG data being applied for this study even if it just highlights the unreliability of IMERG data over Antarctica. Does the IMERG data used in this study have a quality control variable that can be used to mask out unreliable data? From personal experience, I know the IMERG Final Run has this variable. If this is something you didn't consider, I wouldn't recommend rerunning your analysis of the IMERG data, but perhaps just mentioning that the quality control wasn't implemented since this isn't an important aspect of the study.

Figure 2: Can you add date labels to the two panels in this figure?

Line 285: From the sentences that follow, this seems dependent on which station you are describing. Like you say the RCM estimates at Channel are quite accurate in the next sentence.

Line 299-300: Careful with the terminology here. The way this sentence is structured, it sounds like you are making a general statement about RCM performance during the winter and

summer. When addressing your case studies, perhaps it is more intuitive to say, "winter event" and "summer event".

Lines 364-372: I suggest reorganizing this paragraph so that you discuss all the summer event results first and then finish with the couple results on the winter event rainfall or the other way around.

Line 432-433: Perhaps rephrase to "which may indicate that in the most extreme AR conditions during winter, liquid precipitation can still occur". It's not very surprising it rains during summer AR events, but the winter rain is more impressive.

Figure 5: Can you add date ranges to the figure caption?

Figure 6: Similar comment as before, please mention which date the transects are from.

Figure 7: Same as previous comment.

Figure 8: Same, a date range would be helpful in the caption

Section 3.4: This section does an excellent job at covering many different facets of rainfall behavior during the two AR events. There are some details about the influence of the foehn wind on increasing temperatures and altering the precipitation patterns, but it would be helpful to see an organized paragraph discussing the timing of the foehn relative to the rainfall and whether the foehn leads to enhanced sublimation of precipitation. In its current form, it appears the foehn helps warm the surface allowing more rainfall to happen even though one would expect greater sublimation. Are there differences between the RCMs in how they resolve the timing of the foehn and magnitude of sublimation?

Line 473-475: These sentences could be shortened and combined for simplicity. Like "… indicate the maximum height of the 0 C isotherm which can be considered the melt layer". It's obvious that below the melt layer, precipitation would fall as rain or sublimate.

Line 484: Do you mean sublimation instead of evaporation?

Lines 544-547: I feel like this message about "the difficulty of verifying rainfall but that the RCMs are simulating it" is repeated a few times already like in lines 401-404 and 431-433. Especially here this message doesn't need to be repeated but do keep the interesting discussion on liquid precipitation at low temperatures.

Lines 558-559: It might be worth mentioning that ARs are shifting poleward along with the general storm track (Chemke, 2022).

Technical corrections

Line 68: "the precipitation phase" or "precipitation phasing". Also the Gehring et al. (2022) study is perhaps one of the more closely related studies so it could be worthwhile spending another sentence describing how they found that local foehn winds and orography can completely sublimate the precipitation in some AR cases while AR orientation is crucial for controlling snowfall amounts

Line 114: Don't forget about the oxford comma.

Line 115: Do you mean the models have 16, 20, and 14 levels in the 1km nest?

Line 187-188: This sentence repeats itself, "Then, the observed snow height is compared to the observed snow height to determine the accumulation terms."

Figure 3: It could be helpful if you labelled "summer" and "winter" on the figure so that the reader quickly identifies which event is which.

Line 423: "where the terrain beings to slope upwards more strongly" doesn't sound accurate. How about "where the terrain begins increasingly steepening"?

Line 550: "ARs"

Line 568: "showed"

**References:**

Chemke, R. (2022). The future poleward shift of Southern Hemisphere summer mid-latitude

storm tracks stems from ocean coupling. *Nature Communications*, *13*(1), 1730.

https://doi.org/10.1038/s41467-022-29392-4

Vignon, É., Roussel, M.-L., Gorodetskaya, I. V., Genthon, C., & Berne, A. (2021). Present and

Future of Rainfall in Antarctica. *Geophysical Research Letters*, *48*(8), e2020GL092281.

https://doi.org/10.1029/2020GL092281

---

## Referee Comment (RC2)

**General Comments:**

This study presents a comprehensive analysis of extreme precipitation events in West Antarctica using a mini-ensemble of three state-of-the-art kilometer-scale regional climate models (RCMs). The authors effectively demonstrate that high-resolution RCMs are crucial for capturing the complex interactions between atmospheric rivers (ARs) and the region's topography, which significantly influences precipitation patterns and phase transitions. The study provides valuable insights into orographic uplift, latent heat release, and the formation of supercooled liquid precipitation, contributing to a deeper understanding of precipitation dynamics in this climatically sensitive area.

I find this study to be well-executed and interesting, addressing an important gap in climate modeling for polar regions. The work is valuable for its use of high-resolution modeling to explore the drivers and impacts of ARs in West Antarctica. However, some clarifications and details are needed to strengthen the findings. Overall, I would consider these changes to be minor, as the study is already well-structured and offers valuable insights into the dynamics of extreme precipitation events in West Antarctica.

**Specific Comments:**

**Abstract:**
- Line 18-19: Better to quantify the "severely underestimated" and provide more detail on the difference. For example, "In contrast, ERA5 reanalysis data underestimate snowfall by 40-60% compared to in situ measurements, highlighting the limitations of coarse-resolution reanalysis in capturing localized precipitation events."
- Line 24-25: I would clarify what "resolution insensitive" means and elaborate on the significance of this finding. For example, "Our results demonstrate that RCM-simulated snowfall shows limited sensitivity to spatial resolution, while simulated rainfall increases by up to 50% in 1 km resolution models compared to 12 km models, suggesting that fine-scale modeling is essential to capture rainfall variability accurately."

**Introduction:**
- Line 70-72: Better to highlight the mechanisms leading to rainfall. Suggestion: "…This is primarily due to warm air advection and moisture convergence facilitated by ARs, which elevate temperatures and increase the likelihood of liquid precipitation." Also, please add these two recent publications to highlight the AR-driven rainfall events in Antarctica:
  - Bromwich, D. H., and Coauthors, 2024: Winter Targeted Observing Periods during the Year of Polar Prediction in the Southern Hemisphere (YOPP-SH). Bull. Amer. Meteor. Soc., 105, E1662–E1684, https://doi.org/10.1175/BAMS-D-22-0249.1.
  - Bozkurt, D., Carrasco, J., Cordero, R., Fernandoy, F., Gómez, A., Carrillo, B., Guan, B., 2024. Atmospheric river brings warmth and rainfall to the northern Antarctic Peninsula during the mid-austral winter of 2023. Geophysical Research Letters, 51, e2024GL108391, https://doi.org/10.1029/2024GL108391.

- Line 70-7Line 92: I think it is good to specify the dates of case studies or at least to mention summer/winter events.
- Line 91-95: I suggest rewriting the main motivation of the study i) clarifying the research gap, ii) specifying the novelty and relevance, and iii) emphasizing the broader impact in a clearer way. Also, please clarify what is meant by "mini-ensemble.":
  - "This study uses a mini-ensemble of three state-of-the-art kilometer-scale RCMs to explore the interactions between ARs and the TG and PIG ice shelves in the ASE during two extreme precipitation events—a summer and a winter case study. The study aims to quantify the precipitation deposited by ARs, including rain, which can significantly affect the SMB and stability of these ice shelves. Additionally, we evaluate the performance of these high-resolution RCMs in capturing the dynamics of extreme precipitation events in this climatically sensitive region, where in situ observations are limited. By understanding the drivers and impacts of AR-induced extreme precipitation, this work seeks to determine the extent to which ice mass loss from these critical ice shelves and glaciers can be mitigated or exacerbated by changes in accumulation patterns."

**Method:**
- Line 150: Can you please add which version of Polar-WRF was implemented?
- Line 165-170: Which specific variables of ERA5 were used for downscaling? For instance, were SST and SEAICE data provided by ERA5 or another satellite product?
- Line 175-180: Can you please add the temporal resolution of the observational data?
- Line 184-186: Does the SNOWPACK model use directly in-situ measurements in Cavity Camp and Channel Camp described in 2.3? Or any other data source?
- Also, it would be useful if you can label the location of in-situ measurements in Fig. 1 or Fig. 2.

**Results:**
- Line 201: How do you define extreme precipitation in this study? I couldn't find relevant information in Section 2.
- Line 203-205: Interesting to see above 0°C temperatures during the winter case. Do you have any available atmospheric measurements (radiosonde profile)?
- Line 268-273: For the readers, it would be really useful to have a plot showing AR origin, origin and characteristics (e.g., axis, landfall) to better understand these lines as well as discussions in the next paragraph.
- Line 279-283: Following my previous point, can you quantify the impact of the spatial variability in AR landfall locations on the snowfall distribution across the TG ice shelf, particularly between the Cavity and Channel Camp stations? Specifically, how does the timing and trajectory of each AR event correlate with the differences in observed accumulation at these two locations?
- Line 285-301: Given the observed discrepancies in snowfall estimates between SNOWPACK, RCMs, and ERA5, especially in the summer case, can you elaborate on how the representation of AR dynamics (e.g., moisture transport, phase transitions) differs

among the models? How might these differences contribute to the underestimation of snowfall?

- Line 302-310: Is it possible to provide an estimation of the potential uncertainty in the SNOWPACK accumulation totals due to unmeasured processes (e.g., melt, sublimation, wind erosion)? A brief discussion on this would be useful.
- Line 327-328: Please simplify the sentence for clarity. "The lower precipitation estimates from the satellite data, compared to the multi-model mean of the RCMs, likely result from the satellite product's coarse resolution (0.1° × 0.1°, approximately 12 km) and the challenges in detecting precipitation over cold, snow-covered surfaces."
- Line 337-340: Better to break this into two sentences for clarity. "The RCM 5th-95th percentile range suggests that these models capture some of the spatial heterogeneity indicated by differences between Cavity and Channel. However, the time series at grid points corresponding to each station's location (not shown) are very similar, indicating unresolved spatial variations below the 1 km grid scale or inadequately represented processes."
- Line 346-356: Have you considered evaluating the impact of using different temperature thresholds to diagnose rain in SNOWPACK and how this might influence the discrepancies between the snowfall totals estimated by SNOWPACK and the RCMs? A brief discussion on the potential effects of this diagnostic choice could help clarify its implications for your surface mass balance estimates.
- Line 374: Change to "Figures 3a and 3b" for consistency.
- Line 380-382: Please strengthen this by providing a brief explanation of the specific conditions (e.g., exact temperature ranges) that suggest mixed-phase precipitation. This would clarify why mixed precipitation is "likely."
- Line 420 and Fig. 5: Considering that Polar-WRF (Figure 5g) shows more extensive rainfall over the PIG ice shelf than MetUM (Figure 5f), could these differences be linked to how each model simulates the intensity and landfall of ARs over the region?
- Line 465: For clarity and precision, modify to: "temperatures remain below freezing along the TG transect."
- Line 491-497: The observed supercooled liquid precipitation reaching up to 2500 m (Figure 6) suggests unusual atmospheric stability or cloud dynamics. I think this is very interesting, and it would be good to see a detailed vertical profile of temperature, humidity, and wind within these ARs provide further insights.
- Line 496: It is not clear to me what exactly means "This same effect."
- Line 520-540 and Fig. 7: Figure 7 indicates a high rain-to-snow ratio over certain regions, particularly over steep slopes. Is there a possibility that the model is overestimating liquid precipitation due to its handling of mixed-phase clouds or inadequate representation of ice nucleation processes? Could this observation suggest a potential bias in how the model simulation handles phase transitions during AR events?
- Line 585: Rewrite to improve clarity: "The absolute maximum temperatures are also lower, with median air temperatures over the PIG and TG ice shelves approximately 10°C lower in ERA5 than in the RCMs (Table 2)."

**Conclusions:**

- Line 620-664: Considering the computational trade-offs mentioned, what resolution do you propose as the optimal balance between accuracy and efficiency for simulating extreme precipitation events in regions with complex topography like West Antarctica? Could a dynamic resolution approach, where finer grids are applied only in regions of steep topography or during periods of intense precipitation, provide a more efficient modeling strategy?
- Line 620-664: I suggest highlighting the necessity of conducting sensitivity experiments by altering key microphysical parameters, such as cloud droplet number concentrations and ice nucleation rates, to evaluate their impact on precipitation phase transitions. This would help identify which microphysical processes most significantly influence the partitioning between rain and snow in the region.

**Technical Corrections:**

- In several sections, the text refers to complex atmospheric processes (e.g., supercooled liquid precipitation, seeder-feeder mechanisms) without fully explaining their implications or how they are represented in the models. Adding brief explanations or providing context would improve clarity, particularly for readers less familiar with these concepts.
- Line 374: Change "Figure 3b, d" to "Figures 3a and 3b" for consistency.
- Line 386: Use consistent hyphenation: "ground-truthing" or "ground truth."
- Line 550: Correct "Ars" to "ARs."
- Line 606: Remove the extra punctuation after "Abel et al., 2017."

---

## Author Comment (AC1)

General comments

This study takes an in-depth look at how kilometer scale regional climate models (RCMs) resolve the interactions between atmospheric rivers (ARs) and complex topography in a couple case studies centered around the Amundsen Sea Embayment. Their most consequential results indicate that the boost in resolution in these three RCMS (MetUM, Polar-WRF, and HCLIM) leads to the resolution of non-trivial rainfall quantities over the Thwaites and Pine Island ice shelves, especially during the summer AR event. The ERA5, normally considered the gold standard for Antarctic precipitation studies, is unable to depict the rainfall quantities from these AR events. The final experiment of using the MetUM shows that even model resolutions considering high (12 km) are too coarse to simulate topographically influenced rainfall on the Antarctic coastline.

The authors make a great effort using SNOWPACK to calibrate the ground observations for a better comparison with the RCMs and ERA5 instead of just comparing the models directly with the station observations. Despite the consideration, there remains a great deal of uncertainty in validating the performance of the RCMs and ERA5 at this high resolution where the authors meticulously highlight the potential shortcomings of studying this remote region. Other efforts like incorporating IMERG rainfall data and the discussing its shortcomings emphasize the level of detail that went into this study. From my perspective, all the most relevant research was cited and is usefully employed to place the results in context with the existing literature. I believe this study conveys the potential unrealized threat of heavy rainfall on Antarctic coastlines and the need for advanced modeling and observations to further realize this problem. I've made specific and technical comments below thar hopefully help the authors clarify a few details, but I believe this manuscript will be an important contribution to the field after some minor revisions.

-Jonathan Wille

Specific comments

Line 23-25: Can you specify that you used the MetUM for the resolution experiment here?

Revised "RCM-simulated" to "MetUM-simulated"

Line 33-34: Can you elaborate on this mass loss from ice shelves? Like are the ice shelves themselves shrinking in mass or is something else happening? Also please include a reference for Smith et al. (2020)

We have added additional details to this sentence, as follows:

"The ice shelves restraining TG and PIG have been shown to be vulnerable to damage and weakening, and are changing very rapidly**, thinning and retreating in response to oceanic and atmospheric warming** (Lhermitte et al., 2020; Alley et al., 2021)."

And have included the reference for Smith et al – thanks for spotting that it was missing!

Line 39: The wording is a bit confusing here. If accumulation is not compensating for SMB losses on the ice shelves, then perhaps instead of saying "constrain", try "… but snowfall still modulates (or controls) the timing and characteristics of ice shelf collapse, or recession and thinning"

We have revised "constrain" to "quantify"

Line 70-80: This rainfall paragraph is well-written but it could be beneficial to cite (Vignon et al., 2021) which characterized rainfall patterns around Antarctica (i.e. it happens more often than you'd expect) and uses GCMs to show increased rainfall in the future.

Thanks, we had incorporated this study in an older version of the document, but agree that we should include it here. Added the following:

"**Vignon et al. (2021) show that rain falls up to 100 days per year at low elevation regions around the Antarctic coast, including over ice shelves.**"

Line 158: Can you state how deep convection is treated in the HCLIM? This is described for the other models so it would be good to know here as well.

Deep convection is resolved and explicitly represented by the HCLIM's nonhydrostatic dynamics. However, it uses a shallow convection parameterization based on the eddy diffusivity mass-flux framework (de Rooy and Siebesman, 2008; Bengtsson et al. 2017). We have added this information to this paragraph.

Section 2.3: Since this becomes relevant later in the results, can you describe the topography surrounding the stations so that the reader can get a feeling for how orographic lifting might behave here? Also it might be important to state any potential snow measurement issues at these stations like blowing snow?

Thanks for this suggestion. We include mention of the stations' location relative to the slope of the ice stream as follows:

"**The Cavity and Channel stations are located approximately 20 km from the grounding line region of the TG ice stream where the terrain begins to slope steeply upwards.**"

Because we discuss the issues that potentially arise with blowing snow later in the discussion section, we chose not to include it here. We keep this section as a simple description to be consistent with the other descriptive sections in S2.

Section 2.4: Is this method of diagnosing snowfall using SNOWPACK sensitive to blowing snow related errors or would it actually reduce the impacts of blowing snow?

We would argue here that SNOWPACK diagnoses local accumulation, and we then attribute the local accumulation to snowfall. However, if the snow depth measurements are from an area with net convergence of drifting snow mass, the local accumulation would exceed local snowfall, and vice versa. This means that we would not call it a blowing snow related error, but rather that SNOWPACK diagnoses accumulation using the snow depth measurements, rather than snowfall.

We include a brief discussion of the uncertainties and assumptions at the end of section 2.4 as follows:

"**In this study, we compare the local accumulation at the Cavity and Channel stations to modelled snowfall rates. However, it is important to note that the local accumulation can be substantially impacted by net snow erosion if the station is located in a region of drifting snow divergence, or by deposition when the station is in a zone of drifting snow convergence. Other sources of uncertainty concern the density of new snow, which can be in the order of 20-30% (Keenan et al., 2021), and which directly impacts the estimated accumulated mass from snow height measurements. Our approach also assumes that decreases in snow height are correctly captured by the model. Sublimation generally leads to very small changes in snow depth over the period of the described events andmelt during the events was also estimated to be too small to lead to any substantial surface height decrease. The snow depth measurements show a strong increase in snow depth, suggesting that any possible wind erosion of the snow surface was overshadowed by the net increase in mass in the area.**"

Section 3.1: It would be helpful if you gave more details on the temporal evolution of both case studies. Like providing dates for the AR landfalls. Also at the risk of self-advertising, it could also be helpful to state that these events were detected AR events from the Wille et al. (2021) detection algorithm (I already verified this).

Thanks for this suggestion – we think this information strengthens the statement too. We have included more details in this section. We have not included a great deal more information about the temporal evolution of the cases because we feel this has been done at length in Maclennan et al.

Section S2.1: It was instructive to see IMERG data being applied for this study even if it just highlights the unreliability of IMERG data over Antarctica. Does the IMERG data used in this study have a quality control variable that can be used to mask out unreliable data? From personal experience, I know the IMERG Final Run has this variable. If this is something you didn't consider, I wouldn't recommend rerunning your analysis of the IMERG data, but perhaps just mentioning that the quality control wasn't implemented since this isn't an important aspect of the study.

Thanks for this suggestion. We didn't use the QC flag for the IMERG work as it did not prove to be important to the conclusions, but we have added mention that it was not used in the discussion of the data in the supplementary material.

Figure 2: Can you add date labels to the two panels in this figure?

Good idea, we have included date labels.

Line 285: From the sentences that follow, this seems dependent on which station you are describing. Like you say the RCM estimates at Channel are quite accurate in the next sentence.

This assessment is correct. We have revised this sentence to make this clearer, as follows: "In **the** summer **case**, in comparison to SNOWPACK simulated accumulation, all RCMs under-estimate snowfall and do not capture the exact timing of the snowfall related to the ARs (Figure 3a), **although this depends somewhat on the station considered."**

Line 299-300: Careful with the terminology here. The way this sentence is structured, it sounds like you are making a general statement about RCM performance during the winter and summer. When addressing your case studies, perhaps it is more intuitive to say, "winter event" and "summer event".

Thank you, we have revised this paragraph and the previous to make it clearer that we are describing the behaviour only in these cases.

Lines 364-372: I suggest reorganizing this paragraph so that you discuss all the summer event results first and then finish with the couple results on the winter event rainfall or the other way around.

Thank you, we have reorganised to discuss the winter rainfall results first, and then the summer results.

Line 432-433: Perhaps rephrase to "which may indicate that in the most extreme AR conditions during winter, liquid precipitation can still occur". It's not very surprising it rains during summer AR events, but the winter rain is more impressive.

Thank you. Revised to "However, two out of three simulate modest quantities of rain during the winter case, which may indicate that even in winter liquid precipitation can still occur during the most extreme AR conditions."

Figure 5: Can you add date ranges to the figure caption?

Added.

Figure 6: Similar comment as before, please mention which date the transects are from.

Added.

Figure 7: Same as previous comment.

Added.

Figure 8: Same, a date range would be helpful in the caption

Added.

Section 3.4: This section does an excellent job at covering many different facets of rainfall behavior during the two AR events. There are some details about the influence of the foehn wind on increasing temperatures and altering the precipitation patterns, but it would be helpful to see an organized paragraph discussing the timing of the foehn relative to the rainfall and whether the foehn leads to enhanced sublimation of precipitation. In its current form, it appears the foehn helps warm the surface allowing more rainfall to happen even though one would expect greater sublimation. Are there differences between the RCMs in how they resolve the timing of the foehn and magnitude of sublimation?

Thank you, we are glad you found this section useful. We plan to do some further analysis to investigate the timing of the rainfall relative to the foehn event and expand the description of the influence of the foehn on temperature and precipitation patterns during the case.

Line 473-475: These sentences could be shortened and combined for simplicity. Like "… indicate the maximum height of the 0 C isotherm which can be considered the melt layer". It's obvious that below the melt layer, precipitation would fall as rain or sublimate.

We've chosen to keep this in, for clarity of understanding.

Line 484: Do you mean sublimation instead of evaporation?

Evaporation is accurate here, because we are talking about the loss of rain specifically (i.e. the transformation of liquid to the gas phase).

Lines 544-547: I feel like this message about "the difficulty of verifying rainfall but that the RCMs are simulating it" is repeated a few times already like in lines 401-404 and 431-433.

Especially here this message doesn't need to be repeated but do keep the interesting discussion on liquid precipitation at low temperatures.

Thanks, we have removed some repetitive language in this paragraph.

Lines 558-559: It might be worth mentioning that ARs are shifting poleward along with the general storm track (Chemke, 2022).

Thank you, we have included reference to Chemke (2022) here.

Technical corrections

Line 68: "the precipitation phase" or "precipitation phasing". Also the Gehring et al. (2022) study is perhaps one of the more closely related studies so it could be worthwhile spending another

sentence describing how they found that local foehn winds and orography can completely sublimate the precipitation in some AR cases while AR orientation is crucial for controlling snowfall amounts

Thanks, we have included more discussion of Gehring et al:

"**That study also showed how the specifics of how the airflow interacts with steep topography can strongly influence precipitation phase, for example via the impact of foehn on the sublimation of precipitation particles.**"

Line 114: Don't forget about the oxford comma.

Thanks

Line 115: Do you mean the models have 16, 20, and 14 levels in the 1km nest?

We mean that there are this number of levels below 1000 m altitude. We have revised for clarity as follows:

"**including** 16, 20 and 14 levels below 1 km **altitude.**"

Line 187-188: This sentence repeats itself, "Then, the observed snow height is compared to the observed snow height to determine the accumulation terms."

Woops. One of those should be "calculated". Changed.

Figure 3: It could be helpful if you labelled "summer" and "winter" on the figure so that the reader quickly identifies which event is which.

Thanks for this suggestion, implemented.

Line 423: "where the terrain beings to slope upwards more strongly" doesn't sound accurate. How about "where the terrain begins increasingly steepening"?

Revised to "where the terrain begins to slope upwards more steeply"

Line 550: "ARs"

Revised, thanks.

Line 568: "showed"

We use the present text to refer to study findings elsewhere, so we will keep this as "show" here for consistency.

References:

Chemke, R. (2022). The future poleward shift of Southern Hemisphere summer mid-latitude storm tracks stems from ocean coupling. Nature Communications, 13(1), 1730. https://doi.org/10.1038/s41467-022-29392-4

Vignon, É., Roussel, M.-L., Gorodetskaya, I. V., Genthon, C., & Berne, A. (2021). Present and Future of Rainfall in Antarctica. Geophysical Research Letters, 48(8), e2020GL092281. https://doi.org/10.1029/2020GL092281

---

## Author Comment (AC2)

We are grateful to the two reviewers for the time taken to review our manuscript. We are pleased that both reviewers consider this work to be of high quality and importance to the community. We have considered their suggestions and respond to them below, in red.

General Comments:

This study presents a comprehensive analysis of extreme precipitation events in West Antarctica using a mini-ensemble of three state-of-the-art kilometer-scale regional climate models (RCMs). The authors effectively demonstrate that high-resolution RCMs are crucial for capturing the complex interactions between atmospheric rivers (ARs) and the region's topography, which significantly influences precipitation patterns and phase transitions. The study provides valuable insights into orographic uplift, latent heat release, and the formation of supercooled liquid precipitation, contributing to a deeper understanding of precipitation dynamics in this climatically sensitive area.

I find this study to be well-executed and interesting, addressing an important gap in climate modeling for polar regions. The work is valuable for its use of high-resolution modeling to explore the drivers and impacts of ARs in West Antarctica. However, some clarifications and details are needed to strengthen the findings. Overall, I would consider these changes to be minor, as the study is already well-structured and offers valuable insights into the dynamics of extreme precipitation events in West Antarctica.

Specific Comments:

Abstract:

Line 18-19: Better to quantify the "severely underestimated" and provide more detail on the difference. For example, "In contrast, ERA5 reanalysis data underestimate snowfall by 40-60% compared to in situ measurements, highlighting the limitations of coarse resolution reanalysis in capturing localized precipitation events."

Unfortunately given the word limit constraints we cannot add all the words you suggest here, but we have tried to compromise by increasing a quantitative estimate of the underestimation in brackets, as follows:

"By contrast, snowfall estimates from ERA5 reanalysis for both events are severely underestimated **(by 3-4 times)** compared to the measurements."

We have also added more quantitative statements regarding the underestimate in the main text, e.g. at L297:

"Meanwhile, the median cumulative ERA5-derived estimate of total precipitation over TG ice shelf in the summer case is even lower than the mean RCM estimate, at 28 mm w.e**. (3-4 times lower than the SNOWPACK-derived estimates),** although still within the range of the RCM mini-ensemble (Figure 3a). "

Line 24-25: I would clarify what "resolution insensitive" means and elaborate on the significance of this finding. For example, "Our results demonstrate that RCM-simulated snowfall shows limited sensitivity to spatial resolution, while simulated rainfall increases by up to 50% in 1 km resolution models compared to 12 km models, suggesting that fine-scale modeling is essential to capture rainfall variability accurately."

Thanks for this suggestion. As above, we have limited space available in the abstract, but have revised to: "We also show that while the amount of MetUM-simulated snowfall was comparatively resolution insensitive, the amount of rainfall simulated was not, with rainfall amounts **over Thwaites ice shelf 4-16 times** higher in 1 km simulations compared to 12 km simulations."

And included the following in the main text at L623:

"There are few differences in accumulated snowfall amounts between the various resolution domains of the MetUM, **indicating that simulated snowfall totals are comparatively insensitive to resolution**."

And the following at L627:

"Furthermore, while total accumulated snowfall in the 12 km simulation is comparable to 1 km totals, **the accumulated rainfall totals over TG ice shelf are 16 times lower in the 12 km simulation compared to the 1 km simulations during the summer case and 4 times lower during the winter case** (Figure 8). **Across the entire domain**, rain **is** simulated only over the steepest terrain, such as Thurston Island and King Peninsula."

Summer

1.13 mm w.e. 12 km Thwaites ice shelf

17.84 mm w.e. 1 km Thwaites ice shelf

== 16 x higher

Winter

1.44 mm w.e. 12 km Thwaites ice shelf

6.22 mm w.e. 1 km Thwaites ice shelf

= 4.3 x higher

Introduction:

Line 70-72: Better to highlight the mechanisms leading to rainfall.

Suggestion: "…This is primarily due to warm air advection and moisture convergence facilitated by ARs, which elevate temperatures and increase the likelihood of liquid precipitation." Also, please add these two recent publications to highlight the AR-driven rainfall events in Antarctica:

- Bromwich, D. H., and Coauthors, 2024: Winter Targeted Observing Periods during the Year of Polar Prediction in the Southern Hemisphere (YOPP-SH). Bull. Amer. Meteor. Soc., 105, E1662–E1684, https://doi.org/10.1175/BAMS-D-22- 0249.1.
- Bozkurt, D., Carrasco, J., Cordero, R., Fernandoy, F., Gómez, A., Carrillo, B., Guan, B., 2024. Atmospheric river brings warmth and rainfall to the northern Antarctic Peninsula during the mid-austral winter of 2023. Geophysical Research Letters, 51, e2024GL108391, https://doi.org/10.1029/2024GL108391.

Thank you, we have added in your text suggestion and reference to these two new papers. We have also added reference later in the text (L523-524): "**Bromwich et al. (2024) and Bozkurt et al. (2024) also highlight the complexity of processes involved in initiating the phase transition from snow to rain in ARs**."

Line 70-7Line 92: I think it is good to specify the dates of case studies or at least to mention summer/winter events.

Added (L91-92): "This study will use a mini-ensemble of kilometre-scale RCMs to explore how ARs over the ASE interact with the TG and PIG ice shelves in two case studies **(during winter and summer 2020)**…"

Line 91-95: I suggest rewriting the main motivation of the study i) clarifying the research gap, ii) specifying the novelty and relevance, and iii) emphasizing the broader impact in a clearer way. Also, please clarify what is meant by "mini-ensemble.":

o "This study uses a mini-ensemble of three state-of-the-art kilometer-scale RCMs to explore the interactions between ARs and the TG and PIG ice shelves in the ASE during two extreme precipitation events—a summer and a winter case study. The study aims to quantify the precipitation deposited by ARs, including rain, which can significantly affect the SMB and stability of these ice shelves. Additionally, we evaluate the performance of these high-resolution RCMs in capturing the dynamics of extreme precipitation events in this climatically sensitive region, where in situ observations are limited. By understanding the drivers and impacts of AR-induced extreme precipitation, this work seeks to determine the extent to which ice mass loss from these critical ice shelves and glaciers can be mitigated or exacerbated by changes in accumulation patterns."

Thank you for your helpful suggestion. We have revised this paragraph as follows:

"This study uses a mini-ensemble of three state-of-the-art kilometre-scale RCMs to explore the interactions between ARs and the TG and PIG ice shelves during two extreme precipitation events in winter and summer 2020. We aim to quantify the precipitation deposited during the ARs, including rain, which may affect the SMB and stability of these ice shelves. Additionally, we evaluate the performance of these RCMs in this region in capturing the dynamics of extreme precipitation in this climatically sensitive region, where observations are limited. By understanding the drivers and impacts of AR-induced extreme precipitation, this work seeks to determine the extent to which ice mass loss from important ice shelves and glaciers in the region can be modulated by accumulation."

Method:

• Line 150: Can you please add which version of Polar-WRF was implemented?

We used v4.1.1. This detail has been added.

• Line 165-170: Which specific variables of ERA5 were used for downscaling? For instance, were SST and SEAICE data provided by ERA5 or another satellite product?

We used ERA5 for the standard variables for RCM initialisation (T, P, q, winds etc), as well as SST and sea ice. We also use precipitation variables (rain, snow, precip) to compare with the RCMs directly. We have inserted clarification of the variables used at L120-121:

"**We use standard atmospheric variables to initialise the model (temperature, pressure, winds and humidity) as well as sea surface temperature and sea ice**."

• Line 175-180: Can you please add the temporal resolution of the observational data?

Added "…as well as firn temperature, snow height and GPS position **at hourly resolution**." At L181

• Line 184-186: Does the SNOWPACK model use directly in-situ measurements in Cavity Camp and Channel Camp described in 2.3? Or any other data source?

SNOWPACK uses the in-situ observations available, but relies for downwelling long- and shortwave radiation on the reanalysis products. It can also be run using variables from reanalysis, but the simulations we have used here rely on observational measurements.

• Also, it would be useful if you can label the location of in-situ measurements in Fig. 1 or Fig. 2.

Thank you, we have added these into figure 1.

Results:

• Line 201: How do you define extreme precipitation in this study? I couldn't find relevant information in Section 2.

Ordinarily one could define extreme precipitation as being above a threshold (e.g. the 90$^{th}$ percentile) of the climatological average. For instance, Turner et al. (2019) use the "heaviest 10% of daily precipitation amounts" to indicate extreme precip. However, because this observed dataset is so new, it is not long enough to really compute average statistics. We selected the two cases based on the anomalies in the time series – these two were the largest accumulation periods during the dataset. However, they also fall into the top 10$^{th}$ percentile of precipitation for the available data. The summer case was also identified by Maclennan et al (2023) as an extreme precipitation case.

• Line 203-205: Interesting to see above 0°C temperatures during the winter case. Do you have any available atmospheric measurements (radiosonde profile)?

Unfortunately not! This is an extremely remote site and wintertime radiosonde ascents are impossible. The only data that are available are (uncrewed) automatic weather station data and snow height sensor data.

• Line 268-273: For the readers, it would be really useful to have a plot showing AR origin, origin and characteristics (e.g., axis, landfall) to better understand these lines as well as discussions in the next paragraph.

Unfortunately we do not have the relevant variables available to produce a plot such as this (IVT for instance) and the smaller domains are too small to show the origin of the ARs. However, the summer case is evaluated in Maclennan et al. (2023) and those authors present a more in-depth analysis of the event(s).

• Line 279-283: Following my previous point, can you quantify the impact of the spatial variability in AR landfall locations on the snowfall distribution across the TG ice shelf, particularly between the Cavity and Channel Camp stations? Specifically, how does the timing and trajectory of each AR event correlate with the differences in observed accumulation at these two locations?

While the landfall location may impact the wider spatial variability of precipitation across Thwaites, we are not sure this is a satisfactory explanation for the observed differences between the Cavity and Channel stations. Because the stations are so close to one another (just 4 km apart), it is unlikely that the landfall locations are the dominant case of differences between the two stations. Rather, as discussed in the text, it is more probable that highly localised characteristics, for instance the presence of depressions and metre-scale surface texture like sastrugi or ridges, as well as blowing snow etc. are responsible for the observed spatial variability.

• Line 285-301: Given the observed discrepancies in snowfall estimates between SNOWPACK, RCMs, and ERA5, especially in the summer case, can you elaborate on how the representation of AR dynamics (e.g., moisture transport, phase transitions) differs among the models? How might these differences contribute to the underestimation of snowfall?

Thank you for this insightful comment. There are indeed varying parameterisation schemes, model philosophy etc of the models / ERA5 which produce these differences. It is beyond the scope of this study to drill into the exact reason for these differences, but it is certainly useful discussion to include. As such, we have included the following in the paragraph:

"**Differences in simulated precipitation totals and the timing of precipitation events between the RCMs could be related to the way the models parameterise cloud microphysics or differences in their dynamics that determine how, when and where the simulated ARs make landfall.** Meanwhile, the median cumulative ERA5-derived estimate of total precipitation over TG ice shelf in the summer case is even lower than the mean RCM estimate, at 28 mm w.e. **(3-4 times lower than the SNOWPACK-derived estimates)**, although still within the range of the RCM mini-ensemble (Figure 3a). **This discrepancy is likely to be related to the coarser resolution of ERA5 and its orography, which would affect the passage and trajectory of airflow in the simulations, and hence the interactions between the AR and terrain. Convection is also parameterised in ERA5, which could produce different cloud dynamics and lower precipitation totals than the RCMs, which explicitly resolve convection at 1 km resolution.**"

• Line 302-310: Is it possible to provide an estimation of the potential uncertainty in the SNOWPACK accumulation totals due to unmeasured processes (e.g., melt, sublimation, wind erosion)? A brief discussion on this would be useful.

It's difficult to estimate these potential uncertainties. The largest uncertainty in the accumulated mass would be from the density of the accumulated snow and the interpretation of accumulation in terms of snowfall. The latter assumes that there was no net convergence or divergence of drifting snow in the area. Generally speaking, melt and sublimation during these cases are too small to result in substantial height changes, compared to the height changes caused by the snowfall. We have included some discussion of uncertainties in the manuscript at the end of this paragraph as follows:

"**As discussed in Section 2.4, the biggest sources of error in the SNOWPACK analysis probably come from the estimation of new snow density by the model, and the uncertainty related to blowing snow convergence or divergence. Based on our analysis, we expect that other, negative, mass balance components were small during these events**."

• Line 327-328: Please simplify the sentence for clarity. "The lower precipitation estimates from the satellite data, compared to the multi-model mean of the RCMs, likely result from the satellite product's coarse resolution (0.1° × 0.1°, approximately 12 km) and the challenges in detecting precipitation over cold, snow-covered surfaces."

Thank you, we have revised as follows:

"The lower **precipitation estimates derived from satellite data, as compared** to the multi-model mean of the RCM estimates, **is likely related to** the lower resolution of the satellite product (0.1° × 0.1° or approximately 12 km) and may also reflect the difficulty of satellite sensors in measuring precipitation over cold snow and ice surfaces."

• Line 337-340: Better to break this into two sentences for clarity. "The RCM 5th-95$^{th}$ percentile range suggests that these models capture some of the spatial heterogeneity indicated by differences between Cavity and Channel. However, the time series at grid points corresponding to each station's location (not shown) are very similar, indicating unresolved spatial variations below the 1 km grid scale or inadequately represented processes."

Thank you, we have revised as follows:

"The RCM 5$^{th}$-95$^{th}$ percentile range demonstrates that the RCMs can capture the magnitude of spatial heterogeneity implied by the differences between Cavity and Channel. **However,** the time series at grid points corresponding to each station's location (not shown) are extremely similar, suggesting that there are still unresolved spatial variations below the 1 km grid scale of the RCMs, or processes that are incorrectly represented."

• Line 346-356: Have you considered evaluating the impact of using different temperature thresholds to diagnose rain in SNOWPACK and how this might influence the discrepancies between the snowfall totals estimated by SNOWPACK and the RCMs? A brief discussion on the potential effects of this diagnostic choice could help clarify its implications for your surface mass balance estimates.

This is a good point, that the choice of threshold may play a role. We therefore tested varying the threshold between –2°C and +2°C for the implications of producing rainfall in the SNOWPACK simulation for the Cavity and Channel sites. The figure below (Fig R1) illustrates the results we found. For ERA5 during the summer case, a small change in threshold results in a strong increase in prognosed rainfall. However, as discussed in the manuscript, very strong rainfall rates at the AWS sites would be associated with a decrease in the snow depth because rainfall wets snow, increasing its density and darkening the surface. However, an increase in snow depth is generally found during this period. For MERRA-2 and for both models in the winter case, the threshold would need to be lowered below the freezing point to obtain substantial rainfall.

[Figure]

Fig. R1 – Cumulative rainfall amounts associated with various rain/snow air temperature thresholds used in SNOWPACK to diagnose rainfall.

We have briefly summarised the above results in the revised manuscript as follows:

L367: "**Sensitivity experiments (not shown) where the diagnostic threshold was varied between -2°C and + 2°C showed that in the summer case, lower thresholds resulted in considerably higher SNOWPACK-derived rainfall totals. However, during winter and the threshold needed to be lowered to considerably below zero to prognose rainfall.** Note that…"

• Line 374: Change to "Figures 3a and 3b" for consistency

Revised accordingly.

• Line 380-382: Please strengthen this by providing a brief explanation of the specific conditions (e.g., exact temperature ranges) that suggest mixed-phase precipitation. This would clarify why mixed precipitation is "likely."

We have included the exact temperature range observed at the AWS stations as follows:

"Given that the near-surface atmospheric conditions during this case were near the melting point (see Figures 6 and S2), **with warm air overlying the snowpack and observed 2 m temperatures fluctuating between -2°C and +2°C,** it is likely that precipitation was mixed phase."

• Line 420 and Fig. 5: Considering that Polar-WRF (Figure 5g) shows more extensive rainfall over the PIG ice shelf than MetUM (Figure 5f), could these differences be linked to how each model simulates the intensity and landfall of ARs over the region?

This is certainly a possibility. Polar-WRF is a somewhat wetter, warmer model in these cases, so it could also be related to the availability of heat and moisture.

For example, this is clear in Figures R2a to R2d, which show mean synoptic conditions (including mean near-surface temperature) during the summer case in all 3 RCMs and ERA5.

We have added the following at the end of the section:

"The differences between the RCMs in terms of the spatial distribution of rainfall and its intensity may be related to many factors, including the exact trajectory of the simulated ARs, temperature, moisture availability, model orographic interactions and cloud microphysics."

[Figure]

Fig R2. Mean synoptic conditions, averaged during the summer case, for a) MetUM, b) WRF, c) HCLIM and d) ERA5 over the RCM 1 km model domain. Colours show mean near-surface temperature, while arrows show mean 10 m wind direction. A scale vector is given at the top of each panel for reference.

• Line 465: For clarity and precision, modify to: "temperatures remain below freezing along the TG transect."

Thank you for this suggestion, edited accordingly.

• Line 491-497: The observed supercooled liquid precipitation reaching up to 2500 m (Figure 6) suggests unusual atmospheric stability or cloud dynamics. I think this is very interesting, and it would be good to see a detailed vertical profile of temperature, humidity, and wind within these ARs provide further insights.

• Line 496: It is not clear to me what exactly means "This same effect."

The phrase refers to the process described in the preceding sentence. We have added the following clarification: ". This same effect **(snow transitioning to rain as it falls through a warm layer)** has been observed during combined AR / foehn conditions at Davis station in East Antarctica by Gehring et al. (2022)."

• Line 520-540 and Fig. 7: Figure 7 indicates a high rain-to-snow ratio over certain regions, particularly over steep slopes. Is there a possibility that the model is overestimating liquid precipitation due to its handling of mixed-phase clouds or inadequate representation of ice nucleation processes? Could this observation suggest a potential bias in how the model simulation handles phase transitions during AR events?

It's certainly a possibility – although I would note that in most cases, RCMs tend to overestimate **ice** in mixed phase clouds rather than liquid.

• Line 585: Rewrite to improve clarity: "The absolute maximum temperatures are also lower, with median air temperatures over the PIG and TG ice shelves approximately 10°C lower in ERA5 than in the RCMs (Table 2)."

Revised accordingly, thank you.

Conclusions:

• Line 620-664: Considering the computational trade-offs mentioned, what resolution do you propose as the optimal balance between accuracy and efficiency for simulating extreme precipitation events in regions with complex topography like West Antarctica?

Could a dynamic resolution approach, where finer grids are applied only in regions of steep topography or during periods of intense precipitation, provide a more efficient modeling strategy?

Using dynamic resolution, or indeed the prospect of integrating machine learning and/or other statistical techniques with physically based models could offer considerable advantages at a fraction of the computational cost – both are very exciting prospects for important regions like this.

• Line 620-664: I suggest highlighting the necessity of conducting sensitivity experiments by altering key microphysical parameters, such as cloud droplet number concentrations and ice nucleation rates, to evaluate their impact on precipitation phase transitions. This would help identify which microphysical processes most significantly influence the partitioning between rain and snow in the region.

We have added (L661): "**Additional sensitivity experiments, for example exploring the impact of specific microphysical parameters and processes like droplet or ice crystal number, are beyond the**

**scope of this study but would offer insight into the processes governing precipitation phase partitioning during extreme events."**

Technical Corrections:

• In several sections, the text refers to complex atmospheric processes (e.g., supercooled liquid precipitation, seeder-feeder mechanisms) without fully explaining their implications or how they are represented in the models. Adding brief explanations or providing context would improve clarity, particularly for readers less familiar with these concepts.

We have included a more detailed definition of the seeder-feeder process in the introduction (at L81):

**"Seeder-feeder processes involve a higher-level cloud (the 'seeder') precipitating ice crystals into a lower cloud containing liquid droplets (the 'feeder') where they grow via riming or vapour deposition, thus enhancing liquid precipitation (He et al., 2022)."**

• Line 374: Change "Figure 3b, d" to "Figures 3a and 3b" for consistency.

Revised accordingly.

• Line 386: Use consistent hyphenation: "ground-truthing" or "ground truth."

Thank you, we have revised for consistency.

• Line 550: Correct "Ars" to "ARs."

Done, thanks.

• Line 606: Remove the extra punctuation after "Abel et al., 2017."

Done, thanks.